# Identification of Thermoneutral Zone in Sahiwal Zebu Calves in Subtropical Climate of India

**DOI:** 10.3390/ani15131830

**Published:** 2025-06-20

**Authors:** Brijesh Yadav, Anandita Srivastava, Poonam Yadav, Dilip Kumar Swain, Mukul Anand, Sarvajeet Yadav, Arun Kumar Madan

**Affiliations:** Prof. M.D. Pandey Bio-Climatology Laboratory, Department of Veterinary Physiology, College of Veterinary Science and Animal Husbandry, Veterinary University (DUVASU), Mathura 281001, India; anandisri30@gmail.com (A.S.); poo90nam@gmail.com (P.Y.); dilip_swain@yahoo.com (D.K.S.); drmukulanandvet@gmail.com (M.A.); yadavsarvajeet24@gmail.com (S.Y.)

**Keywords:** climate change, heat stress, cold stress, upper critical temperature, lower critical temperature, threshold THI

## Abstract

The thermoneutral zone is the range of temperatures in which animals have minimal environmental stress and have optimum production. The present study was aimed at identifying the thermoneutral zone in Sahiwal zebu calves. The calves were exposed to increasing and decreasing temperatures, and physiological parameters were recorded. The exposure temperatures at which cold stress and heat stress were initiated with respect to different parameters were identified. During decreasing and increasing exposure temperatures, the first change in few physiological parameters was observed at 18.15 and 30.10 °C, respectively. This indicated that no changes in physiological parameters were observed when the calves were exposed between 18.15 and 30.10 °C. On the basis of these results, the thermoneutral zone for zebu calves in the subtropical region of India is between 18 and 30 °C. Therefore, for optimum productivity and health, the Sahiwal calves can be maintained between 18 and 30 °C, and if the environmental temperature reaches beyond this temperature range, managemental tools should be considered to ameliorate the cold/heat stress in order to minimize health and production losses in Sahiwal zebu cattle.

## 1. Introduction

The TNZ refers to the ambient temperature range in which animals experience minimal thermal stress and maintain homeostasis without requiring additional energy expenditure. Maintaining a thermoneutral environment is a prerequisite for optimum productivity in livestock [1]. When the environmental temperature is either higher or lower than the TNZ, it causes a diversion of energies for the maintenance of thermal balance, leading to thermal imbalance and hence culminating in a loss in production performance and compromised health status. When an animal is housed at a temperature lower than the TNZ, it undergoes cold stress, and if housed above the TNZ, it undergoes heat stress. The upper limit of the TNZ is called the UCT, and the lower limit of the TNZ is known as the LCT. Physiologically, the UCT and LCT are the temperatures at which extra-physiological responses are triggered by the animal to maintain thermoneutrality. Heat/cold stress induces a myriad of behavioral and physiological responses in order to influence energy metabolism and thermodynamics to maintain homeostasis. The chronology of responses to either heat [2,3] or cold [4] stress may vary with the intensity and duration of stress exposure. The TNZ has been established in many livestock species based on different parameters, with vast differences based on species, breed, age, and physiological status. It was reported that the TNZ in European Cattle and Indian Cattle is −1 to 10 °C and 10 to 26 °C, respectively [5]. It is also important to note that the LCT for the cattle adapted in the temperate region is lower than the cattle adapted in the tropical/subtropical regions, and similarly, the UCT for the cattle adapted in the tropical/subtropical regions is higher than those adapted in the temperate climate [5]. Beyond establishing the TNZ, it is important to understand the dynamics and chronology of physiological responses that occur outside of this zone. It is well established that temperature and humidity are the two most important components [6] that finally decide the intensity of heat stress and are being expressed as the temperature humidity index (THI). At ambient temperatures below 25 °C, changes in humidity have minimal physiological effects; however, at higher temperatures, humidity significantly exacerbates heat stress. Therefore, determining both the UCT and the corresponding threshold, the THI is critical.

For the identification of the TNZ, critical temperatures, and threshold THI in different livestock species and breeds, researchers have used different animal responses to increasing and decreasing temperatures. The TNZ varies with the geographical location, species, breed, and physiological status of the animal [7]. Our review of the literature suggested that the TNZ was identified based on the physiological (RR, PR, and RT) and production responses (milk production) before the year 2000 [5,7]. In recent years, the UCT and threshold THIs have been identified on the basis of biochemical [2,3,8,9] and endocrine [3] parameters, conception and pregnancy rate [10,11,12], vaginal temperature [13,14], ruminal temperature [15], milk temperature [16], body surface temperature [17,18,19,20], and antioxidative status [3] in addition to production parameters [3,21]. The UCT differs with different heat stress response parameters. However, it has been unequivocally established that RR is the first parameter that exhibits changes in response to heat stress in livestock species [19,22,23,24,25]. With a progression in heat stress, changes in cortisol levels and RT were observed only at very severe heat stress conditions [3,19]. Heat shock proteins (HSPs) are also identified as one of the markers of the stress [23,26] that has been used to establish threshold THIs [3]. The changes in more important health parameters like hemogram [2] and immune response both in terms of leukograms [23] and cytokines [27] during heat stress have also been used to identify threshold THIs.

Calves are more susceptible to cold stress relative to older cattle because they have a larger surface area to mass ratio, resulting in greater heat loss, and a still-developing rumen that produces little heat associated with ruminal fermentation [28]. Cold stress increases maintenance requirements [29], decreases growth performance [30,31] and immunity [32,33], and increases morbidity and mortality rates [34]. Cold stress presents a well-recognized challenge for calves, particularly neonatal ones. The LCT for dairy calves in temperate regions and dairy cattle in subtropical regions was found to be at 10 °C [4,35,36]. There are only a few studies defining the LCT on the basis of only a few markers of cold stress. Sahiwal calves are native to Pakistan and India and belong to the species *Bos indicus* (zebu), which is well adapted to subtropical climates. Sahiwal calves have anatomical and structural body features that help them to regulate body temperature in hot climates. The identification of the UCT, LCT, and TNZ in zebu cattle calves in the Indian subtropical climate has not yet been done. Similarly, the dynamics and chronology of physiological changes with increasing cold and heat stress has not yet been thoroughly investigated. Therefore, in order to formulate comprehensive and precise heat and cold stress amelioration strategies, the chronology of responses with progression of heat and cold stress should be studied. Therefore, the present study was designed to identify the LCT, UCT, and TNZ in Sahiwal zebu cattle calves.

## 2. Materials and Methods

### 2.1. Place of Study

The experiment was carried out in Prof. M. D. Pandey Bio-Climatology laboratory, Department of Veterinary Physiology, College of Veterinary Science and Animal Husbandry, DUVASU, Mathura, Uttar Pradesh, India. Mathura is located at longitude 78° E and latitude 27° N and an altitude of 176 m above mean sea level in the semi-arid region of the country. The average yearly minimum and maximum ambient temperature ranges from 4 to 46 °C. The mean annual relative humidity ranges from 25 to 85%. The annual rainfall in this area ranges from 200 to 400 mm with an inconsistent distribution all over the year.

### 2.2. Animals

The guidelines of The Committee for the Purpose of Control and Supervision of Experiments on Animals (CPCSEA), Government of India, were followed for experimental procedures as approved by the Institutional Animal Ethics Committee (IAEC), Veterinary University, Mathura (Approval No. IAEC-21/26 dated 30 December 2020). The study complies with Animal Research: Reporting in Vivo Experiments (ARRIVE) guidelines, and all methods were performed following the relevant guidelines and regulations. Six apparently healthy calves (age, 8 to 11 months; weight, 120 to 150 kg, at the beginning of the experiment) were housed in the antechamber of the psychrometric chamber. The antechamber has the capacity to accommodate ten adult animals, is well ventilated with pucca floors, and has a provision of individual feeding, watering, sprinklers, and fans. The calves were fed on total mixed ration (TMR) consisting of wheat straw and concentrate mixture in the ratio of 70:30, consisting of 70% wheat straw, 19% soya de-oiled cake, 7% mustard cake, 1.5% urea, 0.63% salt, 1% di-calcium phosphate, 0.20% magnesium oxide, 0.60% rumen buffer, and 0.07% vitamins A, D_3_, and E to meet the predicted nutrient requirements [37] and were offered *ad libitum* water. The chemical composition (% dry matter basis) of the total mixed ratio offered to experimental animals is presented in Table 1. The deworming of all the calves was done before the commencement of the experiment and repeated every three months by the oral administration of a Fenbendazole bolus (Intas Pharmaceuticals Pvt. Ltd., Ahmedabad, India) at 10 mg/kg body weight, and vaccination against foot and mouth disease, hemorrhagic septicemia, and black quarter (Triovac, Indian Immunologicals, Hyderabad, India) was done as per the farm practices.

### 2.3. Experimental Design

The experiment was conducted in two phases in the psychrometric chamber. In the first phase, six Sahiwal calves were kept for six hours every day between 1000 h and 1600 h at 21, 24, 27, 30, 33, 36, and 39 °C (THIs between 67 and 95). In the second phase of the experiment, the same six Sahiwal calves were kept for similar durations, in the same timeframe as in the first phase, at six different decreasing temperatures between 24 and 9 °C with 3 °C interval (THIs between 72.32 and 50.36). At each temperature and THI, the calves were exposed for 10 days. Before the first temperature exposure, the calves were acclimatized for 10 days at 21 °C in the psychrometric chamber. The calves were exposed to increasing temperatures initially in order to elicit heat stress responses. This was followed by a gap of 3 months, during which they were maintained under normal conditions, and no stress was given. Subsequently, the calves were exposed to decreasing temperatures so that cold stress responses could be elicited. The climatological variables in the psychrometric chamber and antechamber during both phases of the experiment are presented in Table 2. On the 10th day, blood sampling was done, and other physiological parameters were recorded at 1500 h, maintaining the calves within the chamber. The two phases of the experiments were planned in such a way that the psychrometric chamber temperature was within the range of approximately ±4 °C of the antechamber temperature so that animals were not exposed to sudden changes in the temperature when entering or leaving the psychrometric chamber. During each stress period, physiological parameters (RT, RR, and pulse rate (PR)) were recorded, while blood sampling was done for hematological (hemogram and leukogram), biochemical (metabolites and liver enzymes), hormonal (cortisol), and molecular marker (HSP70, HSP90, IL-6, and TNFα) analysis. Trained personnel holding master’s degrees in veterinary physiology collected all data to ensure consistency.

The THI was calculated and used as a heat stress indicator [38].THI = (0.8 × T_db_) + (RH/100) (T_db_ − 14.4) + 46.4 (1)
where T_db_ is dry bulb temperature (°C), and RH is relative humidity (%).

### 2.4. Psychrometric Chamber

In the psychrometric chamber, a desired temperature between 5 to 55 °C and relative humidity (RH) between 15 and 75% can be maintained within the narrow limits (±1.0 °C and ±1.0% RH). The psychrometric chamber is steel-built and insulated with polyurethane foam to maintain airtight conditions and to provide effective thermal insulation. The psychrometric chamber has a dimension of 9.0 m × 4.0 m × 3.0 m with facilities of individual tie stalls and feeders for six large adult animals. The chamber is also equipped with temperature, humidity, and carbon dioxide level sensors (Advance Tech Private Limited, Chandigarh, India), which are connected with an automation panel (Model No. PNL-JSP-SZ-90523-CON-1153; Controls Instruments, Delhi, India) located outside the chamber. The required temperature, relative humidity, and carbon dioxide combination is fixed through the automation panel, and it takes 10 to 30 min to achieve the set of climatological variables in the chamber.

### 2.5. Meteorological Variables

The meteorological variables of the psychometric chamber and antechamber during different phases of the experiment are presented in Table 2.

### 2.6. Physiological Observations

The RR was recorded by observing the flank movement for 1 min and was expressed as breaths per minute. The PR was recorded by observing the pulsation of the middle coccygeal artery at the base of the tail and expressed as beats per minute. The RT was recorded by a clinical thermometer (digital thermometer flexible KLF-102, K-life, Gurugram, India) by inserting into the rectum for 1 to 2 min and was measured in degrees Celsius (°C). All the recordings were done by the same person.

### 2.7. Blood Sampling

The blood samples were collected at 1500 h on the 10th day of every temperature exposure from the jugular vein by vein puncture in sodium heparin-coated vacutainers minimizing the handling stress to the animals. In total, 1 mL from each blood sample was subjected to hematological analysis. Part of the blood was centrifuged at 1700× *g* for 30 min in a swing-out centrifuge (R-8C, S. No. JN-34792, Remi, Mumbai, India) to harvest plasma. The blood plasma was stored at −20 °C for biochemical examination.

### 2.8. Hematological Analysis

From each sample, 1 mL of blood was subjected to hematological analysis by an automated hematology analyzer (Model No- MEK6550K, Nihon Kohden, Osaka, Japan). The total leukocyte count (thousand per µL), total erythrocyte count (millions per µL), hemoglobin (gm/dL), packed cell volume (%), lymphocytes (%), and granulocytes (%) were analyzed in the blood sample.

### 2.9. Biochemical Analysis

The total plasma protein, triglyceride, urea and creatinine concentration, and activity of ALT and AST were estimated with a commercially available kit (Span Diagnostics Ltd., Surat, India) using a biochemical analyzer. Bovine-specific ELISA kits supplied by Bioassay Technology Laboratory, Shanghai, China, were used for the estimation of cortisol, HSP70, HSP90, TNFα, and IL6 as per the manufacturer’s instructions. The intra-assay and inter-assay coefficients of variation (CV) for the estimation of all hormones were <8% and <10%, respectively, and were verified individually for each hormone.

### 2.10. Statistical Analysis

The segmented regression analysis was performed using the program SegReg standard version software to determine the breakpoints [39]. The SegReg model is designed to perform a segmented linear regression of one dependent response on one or two independent variables. The segmentation is done by introducing a breakpoint, which can be a discontinuous or broken line. The selection of the breakpoint is based on maximizing the statistical coefficient of explanation and performing the test of significance. The test of significance was determined using ANOVA. The level of significance was set at *p* < 0.05. When the calves were exposed to increasing temperatures/THIs, the obtained breakpoint temperature and THI after segmented regression analysis was assumed to be the UCT/upper threshold THI. Similarly, when the calves were exposed to decreasing temperatures, the obtained breakpoint temperature after segmented regression analysis was assumed to be the LCT/lower threshold THI.

### 2.11. Mathematical Output

The LCT and UCT, thus obtained for each parameter, were arranged linearly on a temperature scale to identify the highest LCT and lowest UCT between which the parameters recorded have not exhibited any breakpoint.

## 3. Results

Based on the segmented regression analysis using temperature (24–9 °C) against the physiological parameters, the breakpoints obtained were considered as the LCT. Figure 1 indicates that the LCT for RR, PR, and RT was 13.65 °C, 15.5 °C, and 12.15 °C, respectively, while the lower threshold THI was 58.92 for PR and RT and 56.73 for RR (Table 3). Similarly, based on the segmented regression analysis using temperature (21–39 °C) and THI against the physiological parameters, the breakpoints obtained were considered as the UCT and upper threshold THI. Figure 1 indicates that the UCT with respect to RR and RT was 30.10 °C; however, for PR no breakpoint was observed, while the upper threshold for THI was 82.35 for RR and RT, and no breakpoint was found for PR (Table 3).

The breakpoints for the LCT and UCT for erythrocytic parameters are presented in Figure 2, and the lower threshold and upper threshold for THI are presented in Table 3. No significant (*p* > 0.05) LCT was observed for the erythrocytic parameters, whereas the UCT for erythrocytic parameters was observed at 30.10 °C (Figure 2). Similarly, the lower threshold THI for erythrocytic parameters was not observed (*p* > 0.05), while the upper threshold THI was found to be 82.35 for all three erythrocytic parameters (Table 3).

The breakpoints for the LCT and UCT, and the lower and upper threshold THIs for leukocytic parameters are presented in Figure 3 and Table 3, respectively. The breakpoints for the LCT with respect to granulocyte %, lymphocyte %, and total leukocyte count were observed at 18.15, 15.15, and 12.30 °C, respectively. No statistically significant (*p* > 0.05) lower threshold THI was observed for lymphocyte %; however, the lower threshold THI for TLC was 54.97 and for granulocyte % was 63.10. The UCT for total leukocyte count was observed at 30.10 °C, whereas granulocyte % and lymphocyte % exhibited the UCT at 33.07 °C. However, the upper threshold THI for TLC was 82.35 and for granulocyte % and lymphocyte % was 86.94.

The LCTs for AST and ALT activity were observed at 12.30 °C and the UCTs for AST and ALT were observed at 30.10 and 34.77 °C (Figure 4), while the lower threshold THIs for AST and ALT activity were observed at 54.75 and 54.97, respectively, and upper threshold THIs at 88.98 and 82.35, respectively (Table 3).

The LCT for cortisol was observed at 9.15 °C, whereas the UCT for cortisol was exhibited at 32.29 °C (Figure 5). Similarly, the lower and upper threshold THIs for cortisol were observed at 54.97 and 84.29, respectively (Table 3).

Figure 6 and Table 3 show that HSP70 did not show statistically significant (*p* > 0.05) breakpoints for either the LCT or UCT. HSP90 exhibited the LCT at 15.15 °C and UCT at 33.07 °C, with the lower threshold THI at 63.32 and upper threshold THI at 86.94 (Table 3).

The breakpoints for the LCT, UCT, lower threshold THI, and upper threshold THI for TNFα and IL6 are presented in Figure 7 and Table 3. No statistically significant (*p* > 0.05) breakpoint was observed for TNFα. The LCT with respect to IL6 was observed at 12.15 °C, whereas its UCT was observed at 32.22 °C. No lower threshold for IL6 was exhibited (*p* > 0.05), while the upper threshold THI was observed at 85.67.

The breakpoints for the LCTs and UCTs for triglyceride and urea are present in Figure 8. The LCTs for triglyceride and urea were observed at 12.15 and 9.15 °C, respectively, whereas both parameters did not show (*p* > 0.05) any UCTs. Similarly, the lower threshold THIs for triglyceride and urea were observed at 67.05 and 54.97, while both parameters did not show (*p* > 0.05) any upper threshold THIs.

To elucidate the temperature range in which minimal changes were observed in various parameters, the LCTs and UCTs observed for different parameters were arranged on temperature scale, and Figure 9 was obtained. The temperature range between the highest LCT and lowest UCT was 18.15 to 30.10 °C, which was considered as the TNZ for Sahiwal zebu calves.

## 4. Discussion

The Indian zebu cattle are known for heat tolerance and disease resistance in addition to their draft capability. Owing to the exacerbating effects of climate change on the animal production system, the scientific exploration of zebu cattle has become inevitable despite its lower production per lactation. However, in the scenario of climate change, zebu cattle may serve as the right choice for sustainable dairy production in tropical and subtropical countries. The most accepted upper critical THI = 72 or less is identified in *Bos taurus* cattle [40] and is not applicable to *Bos indicus* cattle [3]. Cold stress is more deleterious to calves because of their anatomy and physiology and less developed heat production acclimatization mechanisms [28]. Calf mortality is one of the biggest challenges, which is exacerbated by cold stress in an Indian context. The LCT is the temperature at which heat production and heat conservation mechanisms along with other physiological changes begin to occur. The empirical data regarding the UCTs and LCTs and identification of the TNZ is sparse, and if available, it is based on only very few animal responses. Similarly, the chronological evaluation of stress responses in zebu calves has not been reported before. Therefore, the present experiment was designed to identify LCTs and UCTs and hence identify the TNZ for Sahiwal zebu calves.

Based on our previous studies in cattle [2,23,24,25,26,41], the present study was designed incorporating a total of 18 parameters to elucidate the response of cattle calves to increasing temperature exposure (heat stress) and decreasing temperature exposure (cold stress). The results of the physiological parameters depicted that PR was the first one to be affected with decreasing temperature exposure. The PR increased as the temperature decreased from 24 °C to 15.50 °C and thereafter to maintain sufficient blood flow to sustain body temperature [42]. The results also indicated that the RR has not exhibited any major change until 13.65 °C but exhibited a decreasing trend below 13.65 °C, which may suggest an attempt to minimize the evaporative heat loss through respiration. The LCT for RT indicated that behavioral and physiological mechanisms of heat production and heat conservation were unable to maintain body temperature at exposure temperatures lower than 12.15 °C. An increase in PR in cattle [42], decrease in RT in crossbred cattle [43], and decrease in RR in beef cattle [44] were reported during cold stress. Butt et al. reported a decrease in RT at 8 °C exposure temperature [4], whereas a change in RR, PR, and RT was reported at 7.5 °C [43]. In the present study, LCTs for RR, PR, and RT were observed at relatively higher temperatures in cattle calves due to their poorly developed heat production and conservation mechanisms attributed to their younger age and less heat production by fermentation. It also indicated that calves were more prone to cold stress than adult cattle.

RR and RT are the established markers of heat stress [45,46]. RR has been reported to increase first in response to heat stress with a simultaneous increase in PR, which is followed by an increase in RT [3]. But in the present study, the increase in RR and RT was observed at an ambient temperature of 30.10 °C, which was in accordance with other reports [47] where both RR and RT increased at an exposure temperature of 31 °C in crossbred and zebu calves. The upper critical THI in Holstein bull calves for RR, heart rate, and RT were identified at 82.4, 78.3, and 88.1, respectively [19]. In lactating zebu cattle, the alteration in RR and PR was observed at a THI of 78 and 80, respectively, whereas the RT did not change even at a higher THI of 86 [3]. Yadav et al. reported that the RR, PR, and RT altered only at a temperature exposure of 35 °C (THI: 83.81) in dry non-pregnant crossbred cattle in a climate chamber [2], whereas it was reported that with a progressive increase in heat stress, the RR began to increase at a THI of 74 and remained unaffected until a THI of 80, and the RT began to increase only at a THI of 80 in lactating crossbred cattle [9]. The upper critical THIs for RR, PR, and RT were reported at 77 in dry pregnant Holstein cows [48]. The upper critical THIs for RR, heart rate, and RT were reported to be at 70, 72, and 70, respectively, in Holstein lactating cows [40].

In concurrence with previous studies, the present study also revealed that in calves, the upper threshold THI is higher than in lactating animals for RR and RT. The lower metabolic and fermentative heat production and higher heat loss due to larger surface area per kilogram body weight in calves leads to lower heat load, resulting in the late initiation of physiological heat loss mechanisms as compared to lactating animals. In the present study, the upper threshold THIs for both RR and RT were observed at the same THI, which was contrary to the results of that in Holstein bull calves [19], wherein three different upper critical THIs were identified for RR, heart rate, and RT. It is important to note that in most of the climate chamber studies, the upper critical THIs for RR and RT are reported to be the same [2,47] as in the present study, whereas in studies involving the progression of heat stress in a natural environment, the upper critical THIs for RR and RT are reported to be different [3,9,19,40]. Based on observed breakpoints, PR serves as the most reliable indicator of cold stress, whereas both RR and RT are equally effective physiological markers for detecting heat stress in zebu calves.

Hemogram and leukogram have been used by researchers to study heat and cold stress in different livestock species [49]. In the present study, the results indicated that cold stress did not alter the hemogram; however, an increase in both hemoglobin and RBC count was reported in buffalo calves, and it has been opined that the increase in erythrocyte count, hemoglobin %, and PCV was attributable to hemo-concentration due to lower water intake during cold stress [50]. In many studies, heat stress is reported not to influence the hemogram [51,52,53,54]. In the present study, the initial decrease in erythrocytic parameters at 30.10 °C (THI: 82.35) may be attributable to hemo-dilution due to the ad libitum supply of drinking water, which is in accordance with previous studies [55].

The results reveal that in response to cold stress, one of the first responses was an alteration in granulocyte % followed by an alteration in lymphocyte % and total leukocyte count. An alteration in leukogram indicates that immunological responses were also initiated in response to cold stress like heat stress. An increase in granulocyte and lymphocyte ratio was reported at a mean temperature exposure of 8.7 °C in Korean cattle steers [31]. An increase in leukocyte count was reported at a THI of 84 in crossbred cattle [9]. In the present study, an increase in the neutrophil and lymphocyte ratio was observed at a higher level of heat stress (THI: 86.94), which indicated that cell-mediated immunological responses due to heat stress are initiated only later in the chronology of heat stress responses. In the present study, a decrease in number of granulocytes was observed only beyond 33.07 °C, which may be due to an increase in cortisol levels. Many studies have reported a decrease in lymphocyte count during heat stress, whereas many others have reported an increase or no change [56,57,58]. The critical temperature analysis revealed that the granulocyte% is the most sensitive leukogram parameter for detecting cold stress, whereas TLC serves as the superior indicator of heat stress.

In the present study, a sudden increase in AST activity below 12.30 °C (LCT) and a sudden drop in AST activity above 34.77 °C (UCT) indicate that the metabolic activity exhibited changes outside this range. The LCT and UCT with respect to ALT and AST activity indicate that AST activity is more affected by cold stress as compared to ALT activity. A decrease in both ALT and AST activity during cold stress was also reported [31], whereas Butt et al. reported a progressive decrease in AST activity as in the present study and increase in ALT activity [4], contradictory to the present study, during cold stress in crossbred cattle. The results indicated that cold stress increased the hepatic metabolic rate, while heat stress led to a reduction in it. These changes are probably thyroid mediated and require further study. Serum AST activity was reported to increase at 35 °C, whereas ALT activity increased at 40 °C exposure temperatures in crossbred non-pregnant dry cattle and buffalo [2,59]. AST and ALT exhibited a nearly identical lower threshold THI, while the upper critical THI varied, with ALT exhibiting an early response at 82.35, which was markedly higher as compared to the report by Kim et al. [60], where an increase in both ALT and AST activity at a THI of 74 in calves was observed. A starkly higher upper threshold THI in Sahiwal calves as compared to beef calves [60] may be due to breed differences or a better adaptation to heat stress.

Heat and cold stress triggers the hypothalamic–pituitary–adrenal axis, which stimulates the secretion of cortisol and is used as one of the reliable stress markers; however, the threshold level of heat/cold stress for cortisol secretion differs with species, breed, age, and physiological status of the animal [2,3,19,61]. In the present study, an increase in cortisol levels was observed at 9.5 °C; however, Kim et al. reported increased plasma cortisol levels at −4 °C in Holstein cattle (61). Heat stress causes an initial rise in cortisol levels, reaches, a plateau, and then declines [3,58,62]. Neuwirth et al. reported the UCT for cortisol secretion in dairy bull calves at 32.2 °C at 60% relative humidity (THI: 80.6) [63]. In the present study, the chronology of breakpoints with respect to heat stress responses indicated that changes in the secretion level of cortisol are preceded by many physiological responses, including increases in RR and RT, alterations in a few biochemical parameters, and changes in hemogram and TLC. Our results indicated that an increase in the neutrophil to lymphocyte ratio was due to an earlier increase in cortisol levels during temperature exposure. The results also suggested that RR and RT are better markers of heat stress than cortisol based on threshold THIs [64]. Heat shock proteins (HSPs) are used as indicators of heat and cold stress and exhibit a strong correlation with RT in livestock [3,65,66,67]. Several in vitro studies have been carried out to study the effect of temperature on HSPs, and HSP70 and HSP90 were identified as the most consistently and differentially expressed HSPs [64]. In the present study, with progressively decreasing temperature exposure, plasma HSP90 levels increased below 12.5 °C, preceding the rise in cortisol levels, while alterations in the leukogram occurred subsequently to the changes in HSP90. In the present study, no change in HSP70 level was observed with decreasing temperatures between 24 and 9 °C, and similar results have also been reported in response to cold stress in both calves and steers [61]. The results indicated that HSP90 was a more reliable marker of cold stress than HSP70, based on the identified breakpoint, and its increase appeared to be independent of cortisol elevation, serving as an earlier indicator of cold stress compared to cortisol.

There are only a few studies that have identified the upper threshold THI or UCT for HSPs. In lactating zebu cattle, plasma HSP70 and HSP90 levels began to increase at 28 °C (THI: 78) [3], whereas in Holstein cattle, an initial rise in plasma HSP70 followed by a return to normal levels was observed in response to chronic heat stress [65]. In the present study, no upper threshold THI was observed for HSP70; however, the data points indicate that an increase in plasma HSP70 levels was apparent only at a THI of 90. The results indicated that in zebu calves, the cellular level acclimatization responses in terms of expression of HSPs were delayed as compared to adult cattle as in previous studies [3,65]. The increase in plasma HSP70 and HSP90 even after crossing UCT was not very high in the present study as compared to their expression in PBMCs [62]. The results confirm that the elevation in cortisol was earlier than HSP90 during increasing temperature exposure in zebu calves. The results of the present study suggest that there is limited HSP70-mediated thermo-protection in response to heat stress in calves. The wide variation in plasma HSPs levels in response to increasing THIs in individual animals indicated a significant variation in individual thermotolerance of the animals.

Heat and cold stress adversely affects the immune system of livestock and increases susceptibility to diseases [56,68]. In the present study, IL-6 concentration was observed to increase below 12.5 °C, as temperature exposure reduced from 24 °C to 9 °C, whereas no significant change was observed in TNFα levels during this temperature exposure. In cold stress response, in the chronological order with decreasing temperature exposure, alteration in leukogram succeeded changes in the HSP90 and IL-6 levels, and the increase in cortisol was only at the lowest exposure temperature. The results indicated that in response to cold stress, cortisol was not necessary to trigger the alteration in pro-inflammatory cytokine and HSP90 unlike in heat stress. The results also indicated that the cellular mechanisms in response to cold stress might be different than heat stress. In contradiction to the present study, Nonnecke et al. reported that plasma TNFα was decreased during cold stress in pre-weaned calves [69]. Khalilvandi-Behroozyar et al. reported a higher level of both TNFα and IL6 in cold-stressed calves [68]. In the present study, no significant change in TNFα reflects a non-inflammatory nature of cold stress and strong adaptive resilience of zebu calves.

In many heat stress studies in livestock, the levels and expression of pro-inflammatory cytokines TNF-α and IL-6 have been studied and wide variations have been reported [70]. Plasma TNF-α and IL-6 levels were increased in both heat-stressed *Bos taurus* and zebu cattle [71]. In the present study, the UCT at 32.22 °C and upper threshold THI at 85.67 were observed for IL6, and an increasing pattern in TNFα was observed only after a THI of 86; however, no relationship was found with ambient temperature. The increase in IL6 occurred only after increased levels of cortisol. The results of the present study indicated that a THI beyond 86 may indicate immune dysregulation in zebu calves, which may be the result of multiple physiological changes, including an increase in cortisol levels, HSPs, and oxidative stress. Collier et al. also reported that cortisol exerts negative impact on immunity with the help of HSPs during chronic heat stress [72]. Further, our results indicate that cortisol secretion is succeeded by many cellular changes, including the chaperone activation, pro- and anti-inflammatory cytokine production, and nutrient metabolism.

In the present study, alterations in plasma triglyceride and urea concentration during cold exposure indicated that protein and fat metabolism are affected by cold stress. The results indicated that fat metabolism was altered at higher temperature as compared to protein metabolism during cold stress exposure. Similarly, during heat exposure, although no breakpoint for triglyceride and urea was observed, a decreasing trend in triglyceride concentration was observed with increasing ambient temperatures, which indicated changes in fat and protein metabolism in response to heat stress.

In the present study, Table 3 concludes the sensitivity and chronological order of physiological responses with increasing heat and cold stress. In previous sections, we have discussed the chronology of physiological responses and their sensitivity with respect to different parameters separately. Table 3 concludes that the chronological order of physiological responses to heat stress and cold stress are different. On the basis of the break points, RR, RT, TLC, granulocyte %, AST, ALT, cortisol, IL6, and HSP90 were the sensitive parameters for both cold stress and heat stress, whereas PR, triglyceride, and urea were only sensitive to cold stress, and erythrocytic parameters and lymphocyte % were sensitive only to heat stress. It was astonishing to see that unlike in adult cattle, HSP70 and TNFα were found not be sensitive indicators for both cold and heat stress.

On the basis of the review of literature, this is probably the first study that has encompassed multiple parameters to find out the TNZ in zebu calves. It is important to note that determination of TNZ only on the basis of increase or decrease in RT does not suffice the actual definition of TNZ, as several other physiological parameters change before alteration of RT, both at higher and lower environmental temperatures. Thus, the present study provides a more comprehensive and physiologically accurate TNZ as compared to previous studies. Brody reported the comfort zone for European cattle between 1 and 15 °C and for zebu cattle between 10 and 27 °C on the basis of RR, RT, skin temperature, sweating, and heat production [5]. If similar parameters were only considered in the present study, the TNZ would have been between 12 and 30 °C in zebu calves. The TNZ for calves may range between 0 and 26 °C [69,73], which depends on species, breed, age, nutritional status, hair coat, and climatic conditions [35,74]. The TNZ is a function of LCT and UCT. The LCT in Holstein calves is reported to range from 0 to 18 °C [35], whereas the UCT is reported at 26 °C [75]. In the present study, the LCT and UCT both were found to be higher, which indicated that on one hand, zebu calves are tolerant to higher temperatures—probably due to lower fermentative heat production—and on other hand, they are poorly tolerant to cold stress because of their physiological and anatomical limitations.

## 5. Conclusions

This study demonstrates that UCTs and LCTs vary with different stress parameters. The chronology of physiological responses to heat stress and cold stress differ, indicating that alterations in different stress parameters have different mechanisms in response to heat and cold stress. RR, RT, TLC, granulocyte %, AST, ALT, cortisol, IL6, and HSP90 are the sensitive parameters for both cold stress and heat stress, whereas PR, triglyceride, and urea are only sensitive to cold stress, and erythrocytic parameters and lymphocyte % were sensitive only to heat stress. HSP70 and TNFα are found not to be sensitive to both cold and heat stress. During heat stress, cortisol elevation appears to precede immune and metabolic responses, whereas in cold stress, these physiological changes occur before an increase in cortisol is observed. Based on heat stress responses, the UCT for zebu calves is identified at 30.10 °C (THI: 82.35), whereas based on cold stress responses, the LCT for zebu calves is identified at 18.15 °C. Thus, the TNZ for zebu calves can be proposed to be between 18.15 and 30.10 °C. The results of the study suggest that for the optimum productivity of the Sahiwal zebu calves, the environmental temperature should be maintained between 18 and 30 °C; however, further field validation is recommended. Any deviation beyond this environmental temperature range would require management strategies to optimize productivity in Sahiwal zebu calves.

## Figures and Tables

**Figure 1 animals-15-01830-f001:**
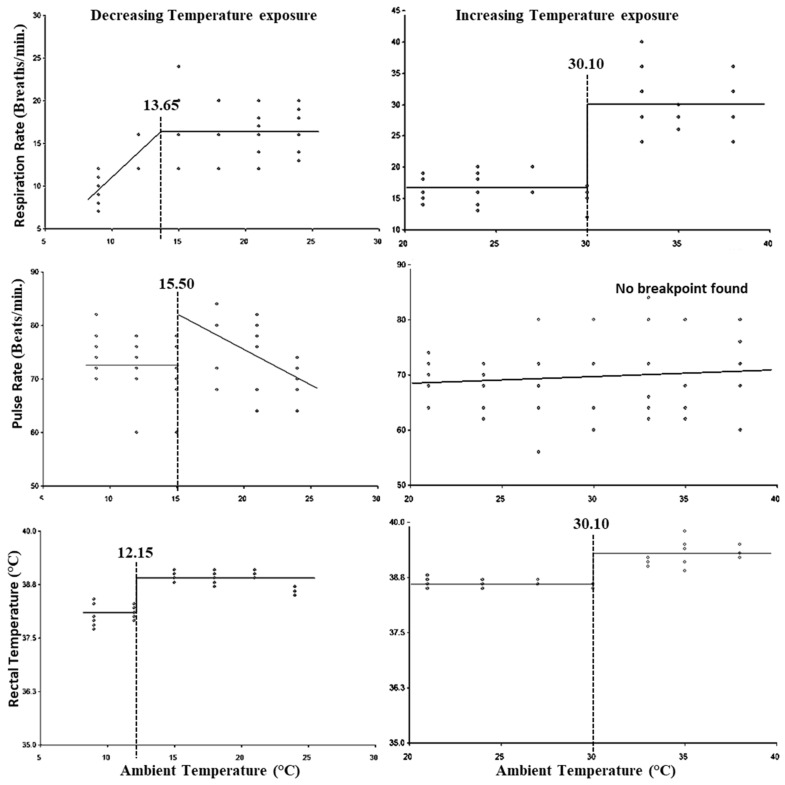
Best fit segmented regression with breakpoints between different physiological parameters and ambient temperature.

**Figure 2 animals-15-01830-f002:**
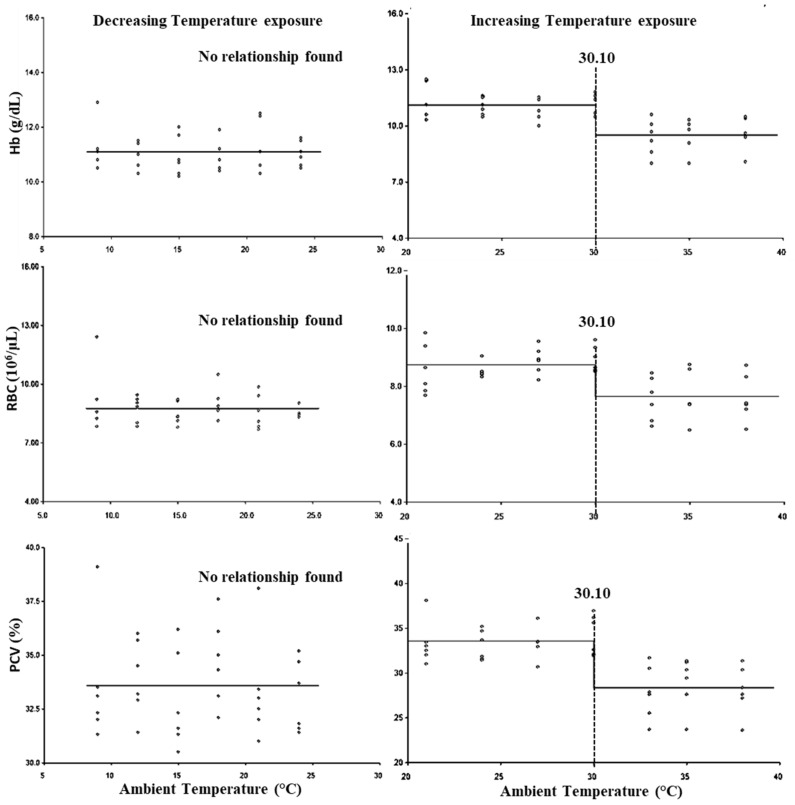
Best fit segmented regression with breakpoints between different erythrocytic parameters and ambient temperature.

**Figure 3 animals-15-01830-f003:**
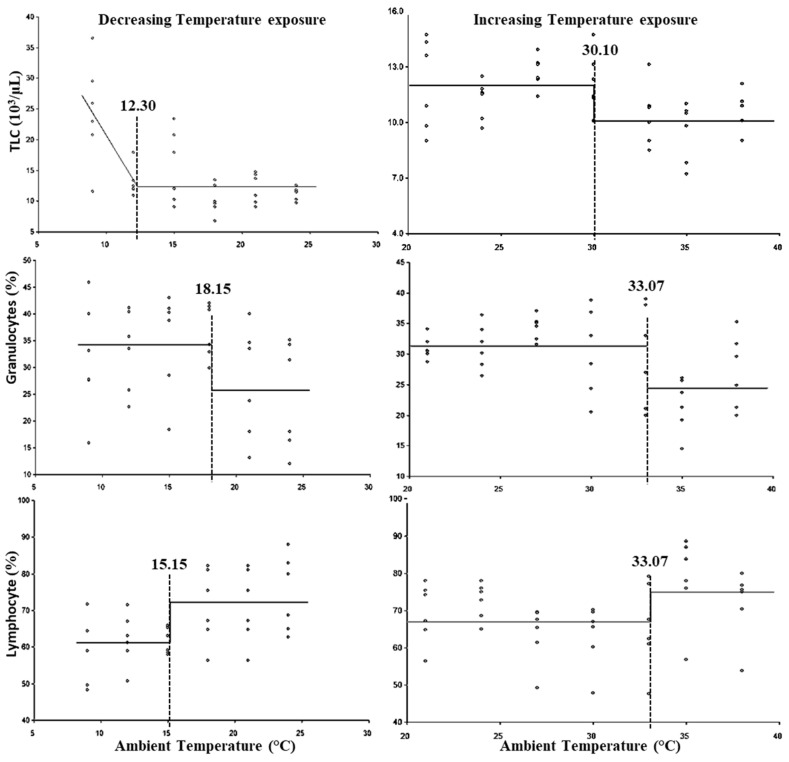
Best fit segmented regression with breakpoints between different leukocytic parameters and ambient temperature.

**Figure 4 animals-15-01830-f004:**
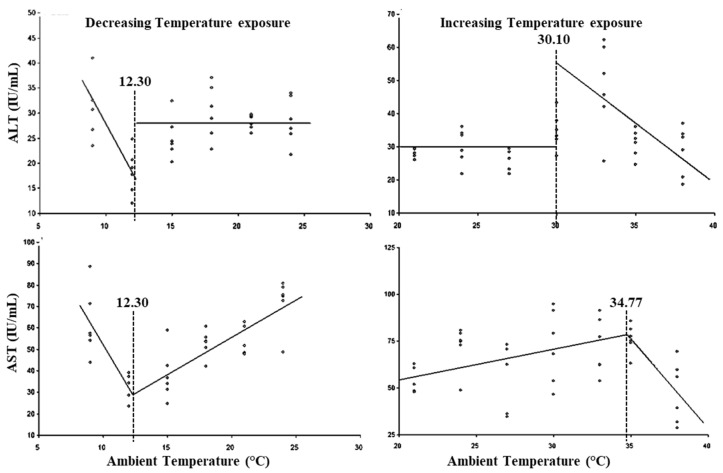
Best fit segmented regression with breakpoints between enzymes and ambient temperature.

**Figure 5 animals-15-01830-f005:**
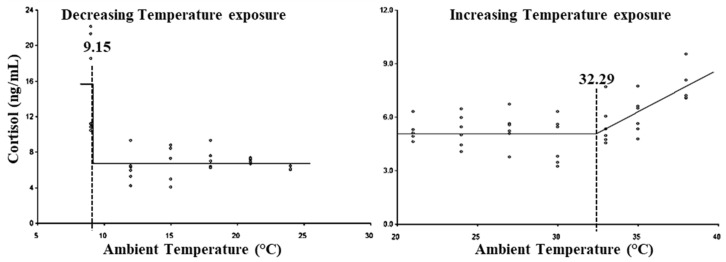
Best fit segmented regression with breakpoints between cortisol and ambient temperature.

**Figure 6 animals-15-01830-f006:**
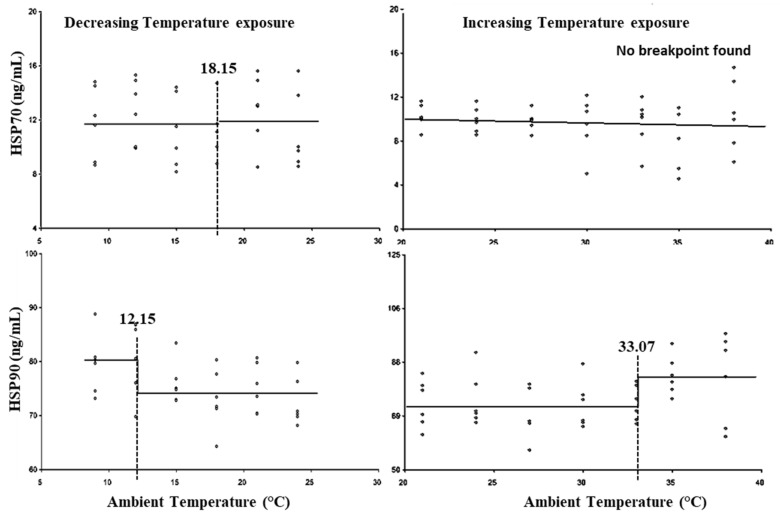
Best fit segmented regression with breakpoints between HSPs and ambient temperature.

**Figure 7 animals-15-01830-f007:**
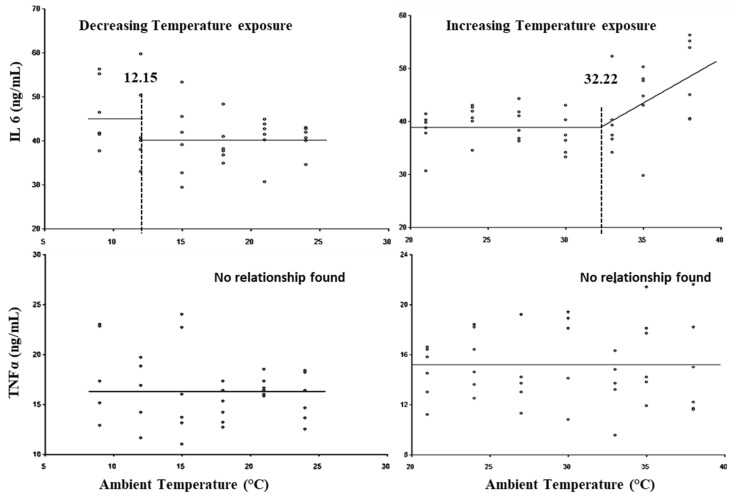
Best fit segmented with breakpoints between cytokines and ambient temperature.

**Figure 8 animals-15-01830-f008:**
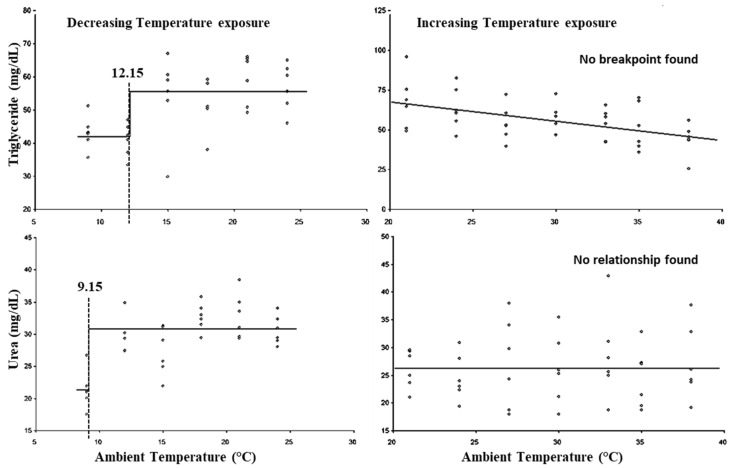
Best fit segmented regression with breakpoints between metabolites and ambient temperature.

**Figure 9 animals-15-01830-f009:**
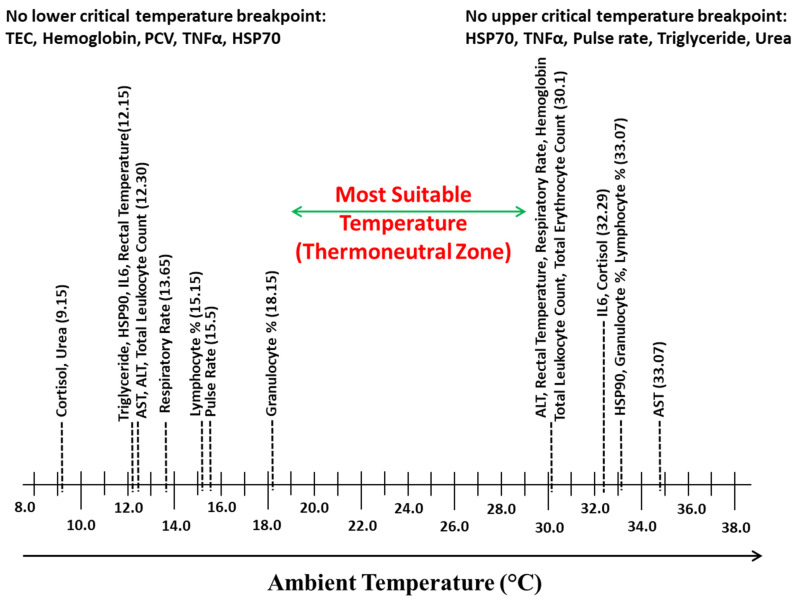
Most suitable temperature (Thermoneutral Zone) identified on the basis of LCT and UCT in Sahiwal zebu calves.

**Table 1 animals-15-01830-t001:** Chemical composition (% dry matter basis) of the total mixed ratio offered to experimental animals.

Attributes	Total Mixed Ration
Organic matter	91.18
Crude protein	17.36
Ether extract	1.46
Neutral detergent fiber	66.73
Acid detergent fiber	17.03
Hemicellulose	20.31
Total carbohydrates	77.86
Calcium	0.87
Phosphorus	0.48

**Table 2 animals-15-01830-t002:** Schedule of temperature (T) and relative humidity (RH) of the ambient and psychrometric chamber.

First Phase						Second Phase					
Psychrometric Chamber			Antechamber			Psychrometric Chamber			Antechamber		
T (°C)	RH (%)	THI	T (°C)		Average RH (%)	T (°C)	RH (%)	THI	T (°C)		Average RH (%)
			Max	Min					Max	Min	
21	60	67.16	26.73	21.31	68.2	24	70	72.32	28.68	22.72	68.6
24	70	72.32	29.68	23.22	65.2	21	60	67.16	26.73	21.31	68.2
27	75	77.45	31.50	24.86	62.1	18	60	62.96	22.42	15.24	70.2
30	75	82.10	33.53	26.38	59.3	15	60	58.76	20.54	13.60	75.2
33	75	86.75	34.82	26.64	75.3	12	60	54.56	19.68	10.82	80.2
36	72	90.75	34.96	27.21	81.4	9	60	50.36	16.54	7.25	82.5
39	70	94.82	35.64	28.32	83.6	--	--	--	--	--	--

**Table 3 animals-15-01830-t003:** The lower and upper critical THIs for different stress parameters in Sahiwal dairy calves.

S. No	Parameter	Lower Threshold THI	Upper Threshold THI
1	Respiratory Rate	56.73	82.35
2	Pulse Rate	58.92	No Breakpoint
3	Rectal Temperature	58.92	82.35
4	Total Red Blood Cells	No Breakpoint	82.35
5	Hemoglobin	No Breakpoint	82.35
6	PCV	No Breakpoint	82.35
7	Total Leukocytic Count	54.97	82.35
8	Granulocyte %	63.10	86.94
9	Lymphocyte %	No Breakpoint	86.94
10	AST	54.75	88.98
11	ALT	54.97	82.35
12	Cortisol	54.97	84.29
13	HSP70	No Breakpoint	No Breakpoint
14	HSP90	58.92	86.94
15	TNFα	No Breakpoint	No Breakpoint
16	IL6	58.92	85.67
17	Triglyceride	58.92	No Breakpoint
18	Urea	54.97	No Breakpoint

## Data Availability

The original contributions presented in this study are included in the article. Further inquiries can be directed to the corresponding author.

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
