# Peer review of "Identification of Thermoneutral Zone in Sahiwal Zebu Calves in Subtropical Climate of India"

_animals, 2025, doi:10.3390/ani15131830_

Round 1
Reviewer 1 Report
Comments and Suggestions for Authors
Major revise
This study focuses on determining the thermal neutral zone (TNZ) of Sahiwal Zebu calves. Cold/heat stress tests were conducted on the same group of calves under different temperature gradients, controlling variables such as humidity and light to reduce environmental interference. The segmented design (heat stress followed by cold stress) conforms to biological logic; The test results cover physiological indicators (rectal temperature, respiratory rate), blood biochemistry (liver enzymes, metabolites), hormones (cortisol), and molecular markers (HSPs, cytokines), comprehensively evaluating stress response and having important theoretical and practical value, providing scientific basis for livestock management in tropical/subtropical regions.The structure of the manuscript is well-organized, and the language is precise. However, several points require clarification and further elaboration for improvement:
Introduction, Why choose Sahiwal Zebu calves for research in thermoneutral zone? Could you tell me about its characteristics and how it differs from Holstein calves?
Line 57, In addition to the THI, there are other indices that reflect the degree of heat stress, such as GTHI, BICT, and ITC, etc.
Line 115-116, how many of the ante-chamber of psychrometric chamber were used in this study? Six calves were housed in one chamber ? What materials is the roof and walls of this chamber made of? Please provide a detailed description of this chamber. Although only six calves, the age and weight of animals were much different which may caused much different response under different temperatures. |
Line 119, how about the composition and nutrition level of the total mixed ration (TMR)?
Line 128, Why is the measurement taken for only six hours each day? It can't reflect the true state of calves throughout the entire day, especially their condition during the night.
Line 129-131, Can you explain more about humidity in different temperatures? How to control the humidity in Psychrometric chamber ?
Line 137-138, Each temperature condition only lasts for 10 days. How long is the acclimation period?
Line 140-142, the psychrometric chamber temperature was within the range of ±4°C of the ante-chamber temperature ? Some of the results in Table 1 are inconsistent with this description.
Line 154-156,All 6 animals were housed in the same chamber without a control group. How to account for the effects of temperature and humidity on the animals?
Line 157-159, Please provide the models of sensors and automation panel.
Line 243-244:Incorrect expression , the UCT of AST is 34.77℃instead of 30.1℃.
Author Response
Reviewer 1
This study focuses on determining the thermal neutral zone (TNZ) of Sahiwal Zebu calves. Cold/heat stress tests were conducted on the same group of calves under different temperature gradients, controlling variables such as humidity and light to reduce environmental interference. The segmented design (heat stress followed by cold stress) conforms to biological logic; The test results cover physiological indicators (rectal temperature, respiratory rate), blood biochemistry (liver enzymes, metabolites), hormones (cortisol), and molecular markers (HSPs, cytokines), comprehensively evaluating stress response and having important theoretical and practical value, providing scientific basis for livestock management in tropical/subtropical regions. The structure of the manuscript is well-organized, and the language is precise. However, several points require clarification and further elaboration for improvement:
Introduction, Why choose Sahiwal Zebu calves for research in thermoneutral zone? Could you tell me about its characteristics and how it differs from Holstein calves?
Authors A description about the Sahiwal Zebu cattle has been inserted in the introduction and also the gap in knowledge has been described. Sahiwal zebu cattle is particularly significant in India and Pakistan, and it differs not only morphologically but physiologically from Holstein cattle, so needs to be investigated.
Line 57, In addition to the THI, there are other indices that reflect the degree of heat stress, such as GTHI, BICT, and ITC, etc.
Authors Yes. I have mentioned THI only as it is one of the most accepted methods to measure environmental heat stress and in present manuscript we are using THI for measurement of heat stress. Inclusion of GTHI/ BICT/ ITC in our study would have taken away the focus.
Line 115-116, how many of the ante-chamber of psychrometric chamber were used in this study? Six calves were housed in one chamber ? What materials is the roof and walls of this chamber made of? Please provide a detailed description of this chamber. Although only six calves, the age and weight of animals were much different which may caused much different response under different temperatures? Authors As mentioned in the manuscript, there is an antechamber near to the psychometric chamber. The antechamber can accommodate 10 adult cattle, whereas, the psychrometric chamber can accommodate 6 adult animals. The psychrometric chamber is steel built and insulated with a puffed material to maintain airtight conditions and to provide effective thermal insulation. (inserted in the manuscript). In the present study, we have used calves aged between 8 and 11 years and have targeted different physiological parameters which normally remain similar. We have observed in the present experiment that changes in the physiological parameters were similar in all the calves at a particular temperature exposure. |
Line 119, how about the composition and nutrition level of the total mixed ration (TMR)?
Authors: The suggestion has been incorporated.
Line 128, Why is the measurement taken for only six hours each day? It can't reflect the true state of calves throughout the entire day, especially their condition during the night.
Authors In the manuscript, we have mentioned that the temperature in the antechamber was approximately ±4 °C as compared to the psychrometric chamber to ensure minimal effects. Moreover, the day temperature and night temperatures can’t be same in natural conditions. We have taken the response of the animals at the end of the day in the psychrometric chamber itself and not outside the chamber to nullify the effect of outside temperature.
Line 129-131, Can you explain more about humidity in different temperatures? How to control the humidity in Psychrometric chamber?
Authors The psychrometric chamber has a humidifier and dehumidifier system attached with a sensor and automation panel. Once humidity is higher than the required humidity, dehumidifier will start working automatically; and vice-versa. This system is controlled by automation panel located outside the chamber.
Line 137-138, Each temperature condition only lasts for 10 days. How long is the acclimation period?
Authors Before the first temperature exposure, the calves were acclimatized for 10 days at 21 °C in the psychrometric chamber. (Inserted in the manuscript).
Line 140-142, the psychrometric chamber temperature was within the range of ±4°C of the ante-chamber temperature? Some of the results in Table 1 are inconsistent with this description.
Authors Agree. We have added the description to remove the inconsistency.
Line 154-156,All 6 animals were housed in the same chamber without a control group. How to account for the effects of temperature and humidity on the animals?
Authors We did not have any control group in this study. We assumed that physiological responses at a temperature exposure of 24°C acts as control.
Line 157-159, Please provide the models of sensors and automation panel.
Authors The required information has been added in the manuscript.
Line 243-244: Incorrect expression, the UCT of AST is 34.77℃ instead of 30.1℃.
Authors Thanks for pointing out. The correction has been done.
Reviewer 2 Report
Comments and Suggestions for Authors
General comments and suggestions
The study investigated a range of body temperatures of Sahiwal Zebu calves to establish the thermoneutral zone. However, the manuscript lacks some sections, such as a simple summary. The Materials and Methods section is not clearly described. The authors did not use the journal’s reference style. I would suggest that the authors review and describe all the sections clearly and include all the sections of a standard article according to the author’s guidelines in the journal. The authors should define all the abbreviations before they use them for the first time.
Specific comments
Line 2: The title doesn’t reflect the study. Please review lines 97 – 98 and streamline the title. It should also be specific to the geographical location/region, as the study findings cannot be generalised.
Simple summary - Line 8: Please include this section.
Abstract - Lines 22 – 31: There are too many abbreviations, some of which haven’t been defined. For instance, what do TEC and TLC stand for?
Introduction
In this section and throughout the manuscript. Please use the Journal’s referencing style (numbered).
Please streamline this section and provide more information about the topic, highlighting the differences between various geographical locations, such as tropical and temperate climates.
Lines 65 – 67: Please review and support the statement with references.
Materials and Methods
Were the studied animals clinically healthy prior to the study? This information is crucial. For example, it would have been useful if the authors had provided the animals’ TPR (Rectal temperature, Pulse rate, and Respiration rate).
Readers would benefit from the researcher's qualifications and experience. Please provide this information, especially for those who were involved in data collection.
Line 123 – 85: Double-check the word “food”.
Line 125: Which recommendations?
Line 126: Experimental design—please review this section and make sure the information flows nicely. For example, the information provided in line 138 about the number of days calves were kept at each temperature could come early in lines 128 – 129.
Lines 132 – 133: Did you expect heat stress at that temperature? What is the normal body temperature of cattle?
Results
Tables showing the value of each parameter recorded at a given temperature would be nice.
Please review all the figures and make sure they are standalone.
Discussion
This section is too long. It would be nice to concentrate more on the main findings.
Conclusions
Please make the main conclusions more precise and supported by the main findings.
Comments on the Quality of English LanguageCheck on the structure, typing errors, and grammar.
Author Response
Reviewer 2
Authors: We sincerely thank the reviewer for his time and expertise, and valuable suggestions.
The study investigated a range of body temperatures of Sahiwal Zebu calves to establish the thermoneutral zone. However, the manuscript lacks some sections, such as a simple summary. The Materials and Methods section is not clearly described. The authors did not use the journal’s reference style. I would suggest that the authors review and describe all the sections clearly and include all the sections of a standard article according to the author’s guidelines in the journal. The authors should define all the abbreviations before they use them for the first time.
Specific comments
Line 2: The title doesn’t reflect the study. Please review lines 97 – 98 and streamline the title. It should also be specific to the geographical location/region, as the study findings cannot be generalised.
Authors: The title has been changed as per the suggestions.
Simple summary - Line 8: Please include this section.
Authors: The summary has been added as suggested.
Abstract - Lines 22 – 31: There are too many abbreviations, some of which haven’t been defined. For instance, what do TEC and TLC stand for?
Authors: All the abbreviations have been defined as suggested.
Introduction
In this section and throughout the manuscript. Please use the Journal’s referencing style (numbered).
Please streamline this section and provide more information about the topic, highlighting the differences between various geographical locations, such as tropical and temperate climates.
Authors: As suggested 3 lines geographical locations and LCT and UCT has been added .
Lines 65 – 67: Please review and support the statement with references.
Authors: As suggested, reference has been added.
Materials and Methods
Were the studied animals clinically healthy prior to the study? This information is crucial. For example, it would have been useful if the authors had provided the animals’ TPR (Rectal temperature, Pulse rate, and Respiration rate).
Readers would benefit from the researcher's qualifications and experience. Please provide this information, especially for those who were involved in data collection.
Authors: The required information has been added.
Line 123 – 85: Double-check the word “food”.
Authors: Thanks for pointing this mistake. Correction has been incorporated.
Line 125: Which recommendations?
Authors: Suitable amendments have been done.
Line 126: Experimental design—please review this section and make sure the information flows nicely. For example, the information provided in line 138 about the number of days calves were kept at each temperature could come early in lines 128 – 129.
Authors: As per the suggestions, the changes have been done.
Lines 132 – 133: Did you expect heat stress at that temperature? What is the normal body temperature of cattle?
Authors: The body temperature of the cattle is around 38 ⁰C. In temperate cattle breeds, heat stress in initiated at 23-25 ⁰C onwards whereas, in tropical cattle breeds heat stress is initiated at 28-30 ⁰C onwards. It is also seen our study.
Results
Tables showing the value of each parameter recorded at a given temperature would be nice.
Authors: This is a nice suggestion but it will be a repetition of the figures therefore, table showing only temperature is not given. Moreover, last figure is based on temperature only.
Please review all the figures and make sure they are standalone.
Authors: Thanks for the suggestion.
Discussion
This section is too long. It would be nice to concentrate more on the main findings.
Authors: This is a very comprehensive research article and requires detailed discussion. However, few changes have been done to reduce the discussion portion.
Conclusions
Please make the main conclusions more precise and supported by the main findings.
Authors: Thanks for the suggestions. Few changes have been incorporated in the conclusion.
Reviewer 3 Report
Comments and Suggestions for Authors
The manuscript presents a significant and understudied subject: the definition of the thermoneutral zone (TNZ) in Zebu calves. The experimental design is comprehensive and encompasses an extensive variety of molecular, hormonal, biochemical, haematological, and physiological parameters. Nevertheless, there are crucial areas in which clarity, scientific rigour, and language require improvement.
The abstract is overly detailed and lacks a clear structure. The background, objectives, methodology, key results, and implications should be clearly described. The take-home message is currently difficult to extract due to the saturation of data points.
Line 36 needs spacing between “(TNZ) is”.
Line 58 needs spacing between “minimaleffects”.
Line 60 needs to be rephrased in academic language.
Line 75, 80, 82, 84 Need spacing between “2022).The” , “2020).Heat”, “2020).The”, “havealsobeen”.
Line 92 needs spacing between “(LCT)for”
Several paragraphs in the Introduction contain multiple claims supported by only one final reference at the end. This weakens the scientific credibility of the text. It is recommended to insert citations more frequently, directly after each specific claim, rather than at the end of a paragraph.
While informative, the introduction lacks a clearly stated research gap. The rationale for this study and how it adds to the existing literature on Zebu calves should be explicitly stated. Consider condensing the literature review and emphasizing novelty.
Line 115 The use of only six calves significantly limits the study. The authors should provide a justification—whether based on ethical considerations, resource limitations, or prior similar studies. Indicate whether any power analysis was performed.
Line 118 needs spacing between “fans.The”
Line 119 states that calves were fed TMR based on NRC (2001), but lacks key details. Please specify the proportions of wheat straw and concentrate in the TMR, frequency and timing of feeding, whether individual intake was monitored or feed was offered uniformly, nutritional composition of the TMR (e.g., DM, CP, ME content).
Line 132 The phrase “so that” (line 132) is informal and should be replaced with more formal academic alternatives such as “in order to”, “to elicit”, or “with the aim of”.
Line 129-135 In the second phase of the study, the same six calves were re-used after a 3-month gap to assess cold stress responses. However, over such a time interval (from ~8 to ~11 months of age to ~11–14 months), physiological maturity and thermoregulatory capacity change substantially. This may confound the interpretation of THI responses.
Please justify why the same animals were used instead of using two parallel groups of calves of similar age. If this was due to constraints, please discuss the potential impact of age-related differences in the response to heat vs. cold stress in the limitations.
Line 137-139 The sentence "blood sampling was done and other physiological parameters were recorded at 1500 h in the psychrometric chamber itself" (line 137) may cause confusion by implying that the chamber itself performed sampling. Please rephrase to clarify that the animals remained inside the chamber during sampling, and not that the chamber conducted the procedure.
Lines 191–202 The use of segmented regression is appropriate, but details about statistical tests for model fit, assumptions, and any other analyses (e.g., multiple comparisons, significance threshold) are missing.
Line 175–176: Centrifugation details lack rotor radius or g-force.
Please specify the model and manufacturer of the haematology analyser (currently only manufacturer is listed), as well as the biochemical analyser used. This is essential for reproducibility and methodological transparency.
Line 210 The phrase “15.5 °C (non-significant)” is vague. Please indicate the statistical threshold and actual p-value if available. For clarity, use standardized phrasing such as “not significant (p > 0.05)” or include exact p-values.
Line 212 Needs spacing between “56.73for”
Lines 223, 232 “table 2” should be in capital letter.
Lines 225-227 Please rephrase to clearly state whether no significant breakpoint was detected and explicitly name the parameters. Consider avoiding vague phrases like “the same trend” and use more precise language.
Line 234 Sentences needs editing.
Line 263 needs spacing between “Table 2.TNFα”
Line 270 “ureais” should be corrected to “urea is” or preferably rephrased as “triglyceride and urea are presented in Figure 8.”
Line 271 needs spacing between “e 8.LCTs”
Line 280 While "wherein" is grammatically correct and means "in which," it's considered archaic and less common in contemporary academic writing.
The phrase “with respect to” is used repeatedly throughout the results section (e.g., lines 250, 265, 270, etc.). While grammatically correct, its excessive use affects readability. Consider rephrasing for variation and conciseness (e.g., “Cortisol exhibited LCT at...” or “LCT for cortisol was observed at...” instead).
Figures and Table 2 While visualizations are abundant and useful, the in-text explanation is insufficient. Authors should better interpret which parameters were most sensitive and why some showed no breakpoint. For Table 2 consider visually differentiating which parameters showed both LCT and UCT.
Line 299 The abbreviation “LCT” is redefined in the Discussion section, although it has already been introduced earlier in the manuscript. To improve conciseness, the full term and parentheses can be omitted here.
Lines 308 has to many spacing between “2022), the”
Line 310 needs spacing between “stress).The”
Line 327 needs spacing between “responsesto”
Line 329 needs spacing between " increasefirst”
The manuscript inconsistently uses both “THI = ...” and “THI: ...” formats (e.g., lines 327–331). For clarity and consistency, please standardize the presentation to a single style, preferably “THI = ...”, throughout the text.
Line 377 needs spacing between “modilutiondue”
Line 377 “ad-lib” was used in at least one instance (e.g., “ad-lib supply of drinking water”), which is incorrect in the context of animal feeding. The proper Latin phrase is “ad libitum”, or optionally the abbreviation “ad lib.” Please revise accordingly throughout the manuscript.
The abbreviations ALT and AST are used earlier in the Results section without prior explanation but are only defined later in the Discussion (line 396). Please ensure that all abbreviations are spelled out at first mention — ideally in the Methods or Results — and used consistently thereafter.
Line 411 needs spacing between “stressincreased”
Line 412 needs spacing between “ratewhile”
Line 450 The sentence needs references.
Line 500 needs spacing between “IL-6levels”
Line 521 The sentence “On the basis of review of literature; this is probably the first study...” incorrectly uses a semicolon. A comma or rephrasing should be used instead, as the first clause is not an independent sentence.
The discussion is overly long and includes excessive background repetition. It should be refocused on comparing current findings with previous studies and highlighting new insights into the physiology of cold/heat stress in Zebu calves.
The conclusion is repetitive and does not emphasize the main takeaways clearly. The TNZ, its practical relevance, and applications should be highlighted more concisely.
The manuscript uses author-year citation style throughout, which does not conform to Animals guidelines. All citations must be converted to numerical format [1], [2], etc., with the reference list ordered by first mention.
Comments on the Quality of English LanguageThe manuscript would be enhanced by moderate English language editing to enhance consistency, clarity, and sentence structure. It is advisable to have someone who is proficient in academic English conduct a thorough review in order to improve the overall readability.
Author Response
Reviewer 3
Authors: We sincerely appreciate your professional capability which has certainly improved the quality of the manuscript. We thank you for your time and expertise.
The manuscript presents a significant and understudied subject: the definition of the thermoneutral zone (TNZ) in Zebu calves. The experimental design is comprehensive and encompasses an extensive variety of molecular, hormonal, biochemical, haematological, and physiological parameters. Nevertheless, there are crucial areas in which clarity, scientific rigour, and language require improvement.
The abstract is overly detailed and lacks a clear structure. The background, objectives, methodology, key results, and implications should be clearly described. The take-home message is currently difficult to extract due to the saturation of data points.
Authors: The abstract has been rewritten taking into account the suggestions.
Line 36 needs spacing between “(TNZ) is”.
Authors: The correction has been incorporated.
Line 58 needs spacing between “minimaleffects”.
Authors: The correction has been incorporated.
Line 60 needs to be rephrased in academic language.
Authors: The correction has been incorporated.
Line 75, 80, 82, 84 Need spacing between “2022).The” , “2020).Heat”, “2020).The”, “havealsobeen”.
Authors: The correction has been incorporated.
Line 92 needs spacing between “(LCT) for”
Authors: The correction has been incorporated.
Several paragraphs in the Introduction contain multiple claims supported by only one final reference at the end. This weakens the scientific credibility of the text. It is recommended to insert citations more frequently, directly after each specific claim, rather than at the end of a paragraph.
Authors: As per suggestion, where ever case may be, references have been rewritten.
While informative, the introduction lacks a clearly stated research gap. The rationale for this study and how it adds to the existing literature on Zebu calves should be explicitly stated. Consider condensing the literature review and emphasizing novelty.
Authors: Thanks for the comment. The research gap has been explicitly added.
Line 115 The use of only six calves significantly limits the study. The authors should provide a justification—whether based on ethical considerations, resource limitations, or prior similar studies. Indicate whether any power analysis was performed.
Authors Thank you for this query. There were no ethical considerations. This study has been carried out in a psychrometric chamber where only 6 animals were can be accommodated at a time. As in psychrometric chamber studies, all the environmental parameters are controlled and there is minimum variation. There are least change of any error due to environmental variations. There are several studies in psychrometric chamber where the 6 animals have been used a brief list is enclosed
Yadav, B., Singh, G., Wankar, A., Dutta, N., Chaturvedi, V.B. and Verma, M.R., 2016. Effect of simulated heat stress on digestibility, methane emission and metabolic adaptability in crossbred cattle. Asian-Australasian Journal of Animal Sciences, 29(11), p.1585.
Wankar, A.K., Singh, G. and Yadav, B., 2021. Effect of temperature x THI on acclimatization in buffaloes subjected to simulated heat stress: physio-metabolic profile, methane emission and nutrient digestibility. Biological Rhythm Research, 52(10), pp.1589-1603.
Yadav, P., Yadav, B., Swain, D.K., Anand, M., Yadav, S. and Madan, A.K., 2021. Differential expression of miRNAs and related mRNAs during heat stress in buffalo heifers. Journal of Thermal Biology, 97, p.102904.
Yadav, B., Yadav, P., Madan, A.K., Anand, M. and Yadav, S., 2025. Effect of heat wave on physiological responses, body surface temperature, heat load, and panting behaviour in buffalo heifers. Theoretical and Applied Climatology, 156(5), pp.1-12.
Line 118 needs spacing between “fans.The”
Authors: Corrections have been incorporated.
Line 119 states that calves were fed TMR based on NRC (2001), but lacks key details. Please specify the proportions of wheat straw and concentrate in the TMR, frequency and timing of feeding, whether individual intake was monitored or feed was offered uniformly, nutritional composition of the TMR (e.g., DM, CP, ME content).
Authors: The suggestions have been incorporated.
Line 132 The phrase “so that” (line 132) is informal and should be replaced with more formal academic alternatives such as “in order to”, “to elicit”, or “with the aim of”.
Authors: The correction has been incorporated.
Line 129-135 In the second phase of the study, the same six calves were re-used after a 3-month gap to assess cold stress responses. However, over such a time interval (from ~8 to ~11 months of age to ~11–14 months), physiological maturity and thermoregulatory capacity change substantially. This may confound the interpretation of THI responses.
Please justify why the same animals were used instead of using two parallel groups of calves of similar age. If this was due to constraints, please discuss the potential impact of age-related differences in the response to heat vs. cold stress in the limitations.
Authors: Thank you for this comment. I agree with your proposition. However, in case of Zebu cattle the growth is slow, so the effect on physiological parameters are minimal, however, inclusion of another group of animals would have led to inserting individual variation.
Line 137-139 The sentence "blood sampling was done and other physiological parameters were recorded at 1500 h in the psychrometric chamber itself" (line 137) may cause confusion by implying that the chamber itself performed sampling. Please rephrase to clarify that the animals remained inside the chamber during sampling, and not that the chamber conducted the procedure.
Authors: Corrections were incorporated as per the suggestions.
Lines 191–202 The use of segmented regression is appropriate, but details about statistical tests for model fit, assumptions, and any other analyses (e.g., multiple comparisons, significance threshold) are missing.
Authors: The link for the software has been provided and the required information has been added, while to avoid redundancy, few lines were deleted.
Added content is as follows.
The SegReg model is designed to perform a segmented linear regression of one dependent response on one or two independent variables. The segmentation is done by introducing a breakpoint which can be discontinuous or broken line. The selection of breakpoint is based on maximizing the statistical coefficient of explanation and performing the test of significance. Test of significance was determined using ANOVA. The level of significance was set at P<0.05.
Line 175–176: Centrifugation details lack rotor radius or g-force.
Authors: Required information has been added.
Please specify the model and manufacturer of the haematology analyser (currently only manufacturer is listed), as well as the biochemical analyser used. This is essential for reproducibility and methodological transparency.
Authors: Required information has been added.
Line 210 The phrase “15.5 °C (non-significant)” is vague. Please indicate the statistical threshold and actual p-value if available. For clarity, use standardized phrasing such as “not significant (p > 0.05)” or include exact p-values.
Authors: Required information has been added.
Line 212 Needs spacing between “56.73for”
Authors: Correction has been incorporated.
Lines 223, 232 “table 2” should be in capital letter.
Authors: Required corrections has been added.
Lines 225-227 Please rephrase to clearly state whether no significant breakpoint was detected and explicitly name the parameters. Consider avoiding vague phrases like “the same trend” and use more precise language.
Authors: The suggestions have been incorporated.
Line 234 Sentences needs editing.
Authors: The sentence has been edited.
Line 263 needs spacing between “Table 2.TNFα”
Authors: Correction has been incorporated.
Line 270 “ureais” should be corrected to “urea is” or preferably rephrased as “triglyceride and urea are presented in Figure 8.”
Authors: Correction has been incorporated.
Line 271 needs spacing between “e 8.LCTs”
Authors: Correction has been incorporated
Line 280 While "wherein" is grammatically correct and means "in which," it's considered archaic and less common in contemporary academic writing.
Authors: The suggestion has been accepted and also incorporated.
The phrase “with respect to” is used repeatedly throughout the results section (e.g., lines 250, 265, 270, etc.). While grammatically correct, its excessive use affects readability. Consider rephrasing for variation and conciseness (e.g., “Cortisol exhibited LCT at...” or “LCT for cortisol was observed at...” instead).
Authors: Thanks for pointing out the repetition. The use of “with respect to” has been drastically reduced in the manuscript.
Figures and Table 2 While visualizations are abundant and useful, the in-text explanation is insufficient. Authors should better interpret which parameters were most sensitive and why some showed no breakpoint. For Table 2 consider visually differentiating which parameters showed both LCT and UCT.
Authors: As per the suggestions, changes has been done in the discussion section related to table 3.
Line 299 The abbreviation “LCT” is redefined in the Discussion section, although it has already been introduced earlier in the manuscript. To improve conciseness, the full term and parentheses can be omitted here.
Authors: Necessary changes has been done throughout the manuscript.
Lines 308 has to many spacing between “2022), the”
Authors
Line 310 needs spacing between “stress).The”
Authors Correction has been incorporated.
Line 327 needs spacing between “responsesto”
Authors Correction has been incorporated.
Line 329 needs spacing between " increasefirst”
Authors Correction has been incorporated.
The manuscript inconsistently uses both “THI = ...” and “THI: ...” formats (e.g., lines 327–331). For clarity and consistency, please standardize the presentation to a single style, preferably “THI = ...”, throughout the text.
Authors: Suggested changes has been done throughout the manuscript.
Line 377 needs spacing between “modilutiondue”
Authors Correction has been incorporated.
Line 377 “ad-lib” was used in at least one instance (e.g., “ad-lib supply of drinking water”), which is incorrect in the context of animal feeding. The proper Latin phrase is “ad libitum”, or optionally the abbreviation “ad lib.” Please revise accordingly throughout the manuscript.
Author Suggested changes has been done throughout the manuscript.
The abbreviations ALT and AST are used earlier in the Results section without prior explanation but are only defined later in the Discussion (line 396). Please ensure that all abbreviations are spelled out at first mention — ideally in the Methods or Results — and used consistently thereafter.
Author Suggested changes has been done throughout the manuscript.
Line 411 needs spacing between “stressincreased”
Authors Correction has been incorporated.
Line 412 needs spacing between “ratewhile”
Authors Correction has been incorporated.
Line 450 The sentence needs references.
Authors A reference has been added.
Line 500 needs spacing between “IL-6levels”
Authors Correction has been incorporated.
Line 521 The sentence “On the basis of review of literature; this is probably the first study...” incorrectly uses a semicolon. A comma or rephrasing should be used instead, as the first clause is not an independent sentence.
Authors Correction has been incorporated.
The discussion is overly long and includes excessive background repetition. It should be refocused on comparing current findings with previous studies and highlighting new insights into the physiology of cold/heat stress in Zebu calves.
Authors: This manuscript is very comprehensive and authors have tried to encompass the related discussions, however, as per suggestion discussion has been truncated.
The conclusion is repetitive and does not emphasize the main takeaways clearly. The TNZ, its practical relevance, and applications should be highlighted more concisely.
Authors The conclusion has been changed as per the suggestions.
The manuscript uses author-year citation style throughout, which does not conform to Animals guidelines. All citations must be converted to numerical format [1], [2], etc., with the reference list ordered by first mention.
Authors The required changes has been done.
Round 2
Reviewer 1 Report
Comments and Suggestions for Authors
Line 163, calves. How long were the animals continuously used, and does the age in days affect the establishment of the thermoneutral zone?
Line 201, Does the blood test require fasting?
Line 283, P>0.05.
Line 313, Verify whether the LCT is 18.05 and confirm its consistency with the chart data.
Line 460, Elaborate on the potential factors contributing to aberrant HSP70 fluctuations.
Line 532-549, The variation in THI depends on both temperature and humidity; thus, focusing solely on temperature in this study’s findings is incomplete. For more details on the effects of humidity and temperature factors on animal performance, please refer to the following references,
1. Habeeb, A. A., Gad, A. E., & Atta, M. A. (2018). Temperature-humidity indices as indicators to heat stress of climatic conditions with relation to production and reproduction of farm animals. International Journal of Biotechnology and Recent Advances, 1(1), 35-50.
2. Li, M., Liang, X., Tang, Z., Hassan, F. U., Li, L., Guo, Y., ... & Yang, C. (2021). Thermal comfort index for lactating water buffaloes under hot and humid climate. Animals, 11(7), 2067.
Author Response
Reviewer 1
Thanks for your suggestions and queries. We have tried our best to answer your queries and comply with your suggestions.
Reviewer Line 163, calves. How long were the animals continuously used, and does the age in days affect the establishment of the thermoneutral zone?
Authors: The calves were of age 8-11 months at the beginning of the experiment. This was done to ensure a consistent age group and weight. The same calves were used again in the second phase to avoid individual variation. It took 7 months to complete both the phases of the experiment. However, in the case of Zebu cattle, the growth is slow, so the effects on physiological parameters are minimal. Zebu cattle come to puberty at 22-36 months. Therefore, in the present study, TNZ will not be significantly affected.
Reviewer Line 201, Does the blood test require fasting?
Authors: No, fasting is not required for blood sampling.
Reviewer Line 283, P>0.05.
Authors: Correction is incorporated.
Reviewer Line 313, Verify whether the LCT is 18.05 and confirm its consistency with the chart data.
Authors: Thank you so much for pointing this out. Corrections have been done. The actual value is 18.15.
Reviewer Line 460, Elaborate on the potential factors contributing to aberrant HSP70 fluctuations.
Authors: We have mentioned in the last two lines (468-470) of the paragraph the reasons for the aberrant HSP70 fluctuations. However, as suggested, we have added one more sentence to elaborate on the same.
Reviewer Line 532-549, The variation in THI depends on both temperature and humidity; thus, focusing solely on temperature in this study’s findings is incomplete. For more details on the effects of humidity and temperature factors on animal performance, please refer to the following references.
- Habeeb, A. A., Gad, A. E., & Atta, M. A. (2018). Temperature-humidity indices as indicators to heat stress of climatic conditions with relation to production and reproduction of farm animals. International Journal of Biotechnology and Recent Advances, 1(1), 35-50.
- Li, M., Liang, X., Tang, Z., Hassan, F. U., Li, L., Guo, Y., ... & Yang, C. (2021). Thermal comfort index for lactating water buffaloes under hot and humid climate. Animals, 11(7), 2067.
Authors: Kindly see table 3; we have given the details of lower critical temperatures and upper critical temperatures. Beginning with the introduction and till the discussion, we have considered the THI and discussed it in depth, but because of our hypothesis, we have concluded only on temperature. Moreover, this manuscript pertains to zebu caves.
The second reference Li et al. (2021) seems not to be apt as it compares GTHI/ BICT/ ITC which is not the focus of our study.
Reviewer 2 Report
Comments and Suggestions for Authors
The authors have addressed all the comments.
Author Response
Reviewer 2
No suggestions. We thank the reviewer for their valuable insight.
Reviewer 3 Report
Comments and Suggestions for Authors
Lines 23–24: Please rephrase “The present study was conducted to identify thermoneutral zone (TNZ) in Sahiwal zebu calves.” to a more formal and informative opening. For example: “This study aimed to determine the thermoneutral zone (TNZ) in Sahiwal Zebu calves under controlled environmental conditions.”
Lines 24–25: “The experiment was conducted in the psychrometric chamber in two phases on six calves aged 8 to 11 months…” – Consider clarifying whether the same animals were used in both phases.
Line 26: “every day between 1000 hours and 1600 hours” – This level of detail may be unnecessary for the abstract. Consider simplifying to: “for six hours per day over 10 consecutive days.”
Lines 27–28: “at six different increasing temperatures at 24, 27, 30, 33, 36, and 39 °C with corresponding temperature humidity index (THIs) between 67 and 93” – Please ensure proper spacing (“67and” should be “67 and”), and consider simplifying by noting the temperature range (e.g., “from 24 °C to 39 °C”) and THI range.
Lines 29–31: Rephrase for clarity and conciseness. Suggested: “In the second phase, the same calves were exposed to decreasing temperatures (24 °C to 9 °C) to assess responses to cold stress.”
Line 39: There is a formatting error: “(THI: (THI: 82.35)”. Please correct the duplicated parentheses.
Lines 40–41: “While few parameters exhibited similar LCT, and UCT/THI threshold, many parameters did not show LCT and UCT.” – This sentence is vague and should be rephrased for clarity. Suggested: “Only a subset of parameters displayed both identifiable LCT and UCT values, while others did not exhibit clear breakpoints.”
Lines 42–45: Please consider listing only the most relevant and sensitive parameters to avoid clutter. You may summarize as: “Physiological (e.g., RR, RT), haematological (TLC, granulocyte %), and biochemical (AST, ALT, cortisol, IL-6, HSP90) parameters were sensitive to both cold and heat stress.”
Lines 49–50: “The results of the study suggest…” – Consider adding a stronger concluding sentence on practical relevance. Suggested: “These findings can inform climate-adaptive housing and management strategies for improving calf welfare and productivity in subtropical environments.”
Lines 55–57: The definition of TNZ is generally correct but would benefit from clearer phrasing. Suggested revision: “TNZ refers to the ambient temperature range in which animals experience minimal thermal stress and maintain homeostasis without requiring additional energy expenditure.”
Line 75: The phrase “Besides knowing the TNZ...” is informal. Suggested revision: “Beyond establishing the TNZ, it is important to understand the physiological responses that occur outside of this zone.”
Lines 77–81: THI is correctly described, but some grammatical adjustments are needed. Suggest: “At ambient temperatures below 25°C, changes in humidity have minimal physiological effects; however, at higher temperatures, humidity significantly exacerbates heat stress. Therefore, determining both the UCT and the corresponding threshold THI is critical.”
Lines 83–87: Repetition of “different permutations and combinations...” is informal.
Line 96: Citations are malformed: “conditions ([3,19]” should be “[3,19]”.
Lines 98–100: “used to identify threshold THIs” is valid but needs more context. What’s the relevance or accuracy of these markers compared to physiological ones?
Line 112: The statement “has not yet been done” is too informal. Suggest: “has not yet been thoroughly investigated.”
Lines 113–114: “The TNZ for zebu cattle calves is not yet identified.” is repetitive. Consider integrating it into the previous sentence.
Line 125: Replace “ambient temperature range from 4 to 46 ºC” with “ambient temperatures range from 4 to 46 ºC annually” for clarity.
Line 131: "Government of India were used for experimental procedures..." → consider rewording to: "were followed for all experimental procedures, with approval from the..."
Line 135: "clinically healthy" – please clarify how health status was determined (e.g., physical examination, hematology?).
Lines 141–142: Please ensure that Table 1 is placed immediately after its first mention in the text, rather than at the end of the document. This improves clarity and helps the reader interpret the methodology more efficiently. Do the same with the Table 2. Nutritional details of the TMR are missing.
Line 141: Typo in "ad -libitum" – should be italicized and corrected to ad libitum.
Lines 149–150: “werekept” should be corrected to “were kept” (space missing). This occurs several times.
Lines 151–153: It’s not clear why the warm phase used 7 temperatures and the cold phase only 6. Please explain the rationale.
Lines 156–158: Rephrase: “to illicit” → should be “to elicit”. Also, the logic for doing heat stress before cold stress should be briefly explained—was there a physiological or logistical reason?
Line 163: “when the caves were in the psychometric chamber itself” → Please correct “caves” to calves and rephrase for clarity: “while the calves remained inside the chamber.”
Line 170: “Students with master’s degree...” – consider rephrasing to emphasize technical training, e.g., “Trained personnel holding master’s degrees in Veterinary Physiology collected all data to ensure consistency.”
Lines 173–174: Present units consistently in formula definitions. E.g., “Tdb – dry bulb temperature (°C); RH – relative humidity (%)”.
Line 176: “within the narrow limits (±1.0)” – clarify units: is it ±1.0 °C and ±1.0% RH?
Line 178: “puffed material” is vague – consider specifying insulation material type (e.g., polyurethane foam insulation).
Consider checking the crude protein value (8.8%) as it seems low for growing calves—please ensure it aligns with NRC recommendations and explain if otherwise.
Line 190: Please clarify how these values were measured (e.g., data logger, sensors).
Line 195: Define how long RR was counted (e.g., breaths per minute observed over 30 or 60 seconds?).
Line 197–198: Consider mentioning whether thermometers were calibrated, and whether measurements were done by the same person to reduce inter-observer variation.
Line 202: “least stress to the animals” – suggest rephrasing to “minimizing handling stress.”
Line 204: The centrifugation duration and g-force are adequate, but mention the rotor model or radius if available for reproducibility.
Line 217: Clearly state if intra- and inter-assay CVs were verified for each assay in this study.
Line 219: The URL should not be in the middle of the sentence; cite the software properly with a reference.
The re-use of the same animals after 3 months introduces potential age-related confounding which is not adequately addressed.
Blood sampling time need to be clearly defined.
Lines 236–239 Please clarify whether the breakpoint for pulse rate (PR) was included in the final interpretation of LCT, as it was stated to be “non-significant (p > 0.05)”. It is confusing to include it alongside significant breakpoints. You may consider moving it to a separate sentence or explicitly stating it was excluded from defining TNZ.
Line 240 Please correct the spacing in “56.73for RR” — a space is needed between the number and the word.
Lines 241–244 The phrasing “based on the segmented regression analysis using temperature (21–38°C)” is unclear as 21°C was previously part of the cold exposure phase. Confirm whether this range applies only to the heat stress phase. Additionally, specify why no breakpoint was found for PR.
Line 246 and Figure 1 Ensure Figure 1 includes clear axes labels, units. Indicate if p-values for each breakpoint are displayed.
Lines 249–254, Figure 2 please explicitly state that no significant LCT was observed for the erythrocytic parameters (p > 0.05), rather than just saying “not observed”. The term “could not be found” is vague.
Lines 259–266, Figure 3 Avoid repeating phrases such as “lower threshold THI for lymphocyte % could not be deciphered (p > 0.05)”. Consider rephrasing to: “No statistically significant lower threshold THI was observed for lymphocyte % (p > 0.05)”.
Lines 269–273, Figure 4 The results for liver enzymes should include actual breakpoint temperatures and THI values in the text for clarity, rather than relying solely on figures and tables.
Lines 274–278, Figure 5 Sentence construction is unclear: “Similarly, the lower threshold THI was observed at 54.97 and upper threshold THI at 84.29” — please clarify that these values refer to cortisol.
Lines 280–285, Figure 6 Please correct formatting errors such as “(p>00.05)” and provide a clearer sentence structure: e.g., “HSP70 did not show statistically significant breakpoints for either LCT or UCT (p > 0.05).”
Lines 290–295, Figure 7 Rephrase “did not exhibit (p>0.05) any break point” to a more formal alternative: “no statistically significant breakpoint was observed for TNFα (p > 0.05)”.
Line 298–303, Figure 8 The expression “did not show (p<0.05) any UCT” is contradictory. If p < 0.05, a significant result should be present. Confirm and correct — likely it was meant to say “p > 0.05”.
Lines 324–333: Clarify the novelty of the study earlier in the paragraph. Consider stating more directly what is lacking in the literature (e.g., “Chronological evaluation of stress responses in zebu calves has not been reported before.”)
Lines 336–356: The physiological interpretations are reasonable but tend to be speculative in places. Phrases like “in order to maintain sufficient blood flow” and “minimize evaporative heat loss” require stronger citation support or should be softened (e.g., “may suggest an attempt to…”). Also, avoid repetition from the Results.
Lines 370–379: It would be helpful to elaborate why the THI thresholds are reportedly higher in calves than in lactating cows. Is this purely due to metabolic heat production differences?
Lines 390–401: Please clarify if the neutrophil:lymphocyte ratio is explicitly calculated or inferred. The sentence “could have been triggered by increase in cortisol” is speculative; consider rephrasing or supporting with data correlation.
Lines 402–423: This section discussing AST and ALT is lengthy and includes repeated information. Streamline to avoid repeating “increased at XX °C and then declined.” Clearly distinguish what is novel here.
Lines 425–440: Provide a clearer conclusion regarding cortisol’s role in early vs. late stress response. Consider adding a sentence to summarize its diagnostic value relative to other markers like RR or HSPs.
Lines 441–453: The HSP70 and HSP90 comparison is important, but needs clearer structuring. Present the cold stress findings separately from heat stress findings. Also clarify: how was “better marker” assessed — through earlier breakpoint, magnitude of change, or significance?
Lines 455–471: The discussion of variability in HSP expression is relevant but too dense. Condense and emphasize only the most important interpretation points (e.g., HSP90 increased before cortisol in cold stress).
Lines 472–484: Clearly state why TNFα might not respond to cold stress in this context. The current discussion is speculative and somewhat unclear.
Lines 485–493: Consider reducing citations within one sentence. In Line 491, the claim that THI >86 leads to “compromised immune response” needs clearer evidence or a more cautious statement (e.g., “may indicate immune dysregulation”).
Lines 505–513: This is an important section summarizing the key stress markers. Clarify how sensitivity was defined (e.g., lowest breakpoint, highest magnitude of change, etc.).
Lines 514–530: This concluding section nicely highlights the novelty of the study, but consider reducing repetition (especially of TNZ ranges). Clarify that your findings provide a multi-parameter TNZ definition, which is a strength.
Lines 532–533: Please rephrase for clarity. Instead of “On the basis of present study, it can be concluded that…,” use a more concise and active structure, e.g., “This study demonstrates that…”
Line 539: “Were found not be sensitive” is grammatically incorrect. Please correct to “were found not to be sensitive…”
Lines 540–541: This comparative sentence is insightful but should be clarified. Suggest rephrasing to: “During heat stress, cortisol elevation appears to precede immune and metabolic responses, whereas in cold stress, these physiological changes occur before an increase in cortisol is observed.”
Lines 542–544: Consider replacing “approximately” with specific decimal values already reported in the Results for LCT and UCT. This would strengthen the precision of your conclusion.
Lines 545–548: The final management recommendation is appropriate but would benefit from a more cautious tone. Consider adding: “These findings may help inform thermal comfort and housing strategies in calf management; however, further field validation is recommended.”
Use consistent tense (preferably present simple or present perfect when referring to study findings).
Comments on the Quality of English LanguageThe manuscript contains numerous grammatical errors, missing spaces, awkward phrasings, and inconsistent terminology. While understandable, the English requires thorough editing to meet publication standards.
Author Response
Reviewer 3
I am extremely thankful for in depth review of this manuscript and even pointing out very minute formatting mistakes. Your review has brought significant change in the manuscript both technically and linguistically.
Reviewer Lines 23–24: Please rephrase “The present study was conducted to identify thermoneutral zone (TNZ) in Sahiwal zebu calves.” to a more formal and informative opening. For example: “This study aimed to determine the thermoneutral zone (TNZ) in Sahiwal Zebu calves under controlled environmental conditions.”
Authors: Suggestion has been incorporated.
Reviewer Lines 24–25: “The experiment was conducted in the psychrometric chamber in two phases on six calves aged 8 to 11 months…” – Consider clarifying whether the same animals were used in both phases.
Authors: The calves were of age 8-11 months at the beginning of the experiment. This was done to ensure a consistent age group and weight. The same calves were used again in second phase to avoid individual variation. It has been clarified later in the abstract itself while describing about the second phase.
Reviewer Line 26: “every day between 1000 hours and 1600 hours” – This level of detail may be unnecessary for the abstract. Consider simplifying to: “for six hours per day over 10 consecutive days.”
Authors: Suggestion has been incorporated.
Reviewer Lines 27–28: “at six different increasing temperatures at 24, 27, 30, 33, 36, and 39 °C with corresponding temperature humidity index (THIs) between 67 and 93” – Please ensure proper spacing (“67and” should be “67 and”), and consider simplifying by noting the temperature range (e.g., “from 24 °C to 39 °C”) and THI range.
Authors: Suggestion has been incorporated.
Reviewer Lines 29–31: Rephrase for clarity and conciseness. Suggested: “In the second phase, the same calves were exposed to decreasing temperatures (24 °C to 9 °C) to assess responses to cold stress.”
Authors: Suggestion has been incorporated.
Reviewer Line 39: There is a formatting error: “(THI: (THI: 82.35)”. Please correct the duplicated parentheses.
Authors: Suggestion has been incorporated.
Reviewer Lines 40–41: “While few parameters exhibited similar LCT, and UCT/THI threshold, many parameters did not show LCT and UCT.” – This sentence is vague and should be rephrased for clarity. Suggested: “Only a subset of parameters displayed both identifiable LCT and UCT values, while others did not exhibit clear breakpoints.”
Authors: Suggestion has been incorporated.
Reviewer Lines 42–45: Please consider listing only the most relevant and sensitive parameters to avoid clutter. You may summarize as: “Physiological (e.g., RR, RT), haematological (TLC, granulocyte %), and biochemical (AST, ALT, cortisol, IL-6, HSP90) parameters were sensitive to both cold and heat stress.”
Authors: Thanks for your suggestions. However, with your suggestions the use of abbreviations would increase which authors wish to avoid. Therefore, it would be better to go with same sentence.
Reviewer Lines 49–50: “The results of the study suggest…” – Consider adding a stronger concluding sentence on practical relevance. Suggested: “These findings can inform climate-adaptive housing and management strategies for improving calf welfare and productivity in subtropical environments.”
Authors: Suggestion has been incorporated.
Reviewer Lines 55–57: The definition of TNZ is generally correct but would benefit from clearer phrasing. Suggested revision: “TNZ refers to the ambient temperature range in which animals experience minimal thermal stress and maintain homeostasis without requiring additional energy expenditure.”
Authors: Suggestion has been incorporated.
Reviewer Line 75: The phrase “Besides knowing the TNZ...” is informal. Suggested revision: “Beyond establishing the TNZ, it is important to understand the physiological responses that occur outside of this zone.”
Authors: Suggestion has been incorporated with some modifications.
Reviewer Lines 77–81: THI is correctly described, but some grammatical adjustments are needed. Suggest: “At ambient temperatures below 25°C, changes in humidity have minimal physiological effects; however, at higher temperatures, humidity significantly exacerbates heat stress. Therefore, determining both the UCT and the corresponding threshold THI is critical.”
Authors: Suggestion has been incorporated.
Reviewer Lines 83–87: Repetition of “different permutations and combinations...” is informal.
Authors: A part of the sentence is removed.
Reviewer Line 96: Citations are malformed: “conditions ([3,19]” should be “[3,19]”.
Authors: Correction has been incorporated.
Reviewer Lines 98–100: “used to identify threshold THIs” is valid but needs more context. What’s the relevance or accuracy of these markers compared to physiological ones?
Authors: the highlighted part has been inserted in the said sentence. The changes in more important health parameters like hemogram [2], immune response both in terms of leucograms [23] and cytokines [27] during heat stress have also been used to identify threshold THIs
Reviewer Line 112: The statement “has not yet been done” is too informal. Suggest: “has not yet been thoroughly investigated.”
Authors: Suggestions has been incorporated.
Reviewer Lines 113–114: “The TNZ for zebu cattle calves is not yet identified.” is repetitive. Consider integrating it into the previous sentence.
Authors: Suggestions has been incorporated.
Reviewer Line 125: Replace “ambient temperature range from 4 to 46 ºC” with “ambient temperatures range from 4 to 46 ºC annually” for clarity.
Authors: Yearly word has been used for annually
The average yearly minimum and maximum ambient temperature range from 4 to 46 ºC.
Reviewer Line 131: "Government of India were used for experimental procedures..." → consider rewording to: "were followed for all experimental procedures, with approval from the..."
Authors: Suggestions has been incorporated.
Reviewer Line 135: "clinically healthy" – please clarify how health status was determined (e.g., physical examination, hematology?).
Authors: The word apparently healthy would be more appropriate as the animals were selected on physical examination basis.
Reviewer Lines 141–142: Please ensure that Table 1 is placed immediately after its first mention in the text, rather than at the end of the document. This improves clarity and helps the reader interpret the methodology more efficiently. Do the same with the Table 2. Nutritional details of the TMR are missing.
Authors: It has been in added in the revised manuscript.
Authors: The place for Table 1 has been changed where for Table 2, it is placed just where it is mentioned in the text.
Reviewer Line 141: Typo in "ad -libitum" – should be italicized and corrected to ad libitum.
Authors: Suggestions has been incorporated.
Reviewer Lines 149–150: “werekept” should be corrected to “were kept” (space missing). This occurs several times.
Authors: Suggestions has been incorporated.
Reviewer Lines 151–153: It’s not clear why the warm phase used 7 temperatures and the cold phase only 6. Please explain the rationale.
In the original experimental design, 7 temperatures were taken, but during experiment when we saw that at 15 ⁰C, pilo-erection was observed and slight shivering was observed, whereas at 12 ⁰C excessive shivering was observed. Similarly we also observed decrease in rectal temperature at 12 ⁰C. Then, it was decided that we shall proceed for one lower temperature (9 ⁰C). Even going lower at 6 ⁰C could have caused severe health issues which was avoided.
Reviewer Lines 156–158: Rephrase: “to illicit” → should be “to elicit”. Also, the logic for doing heat stress before cold stress should be briefly explained—was there a physiological or logistical reason?
Authors: Suggestions has been incorporated.
It could have been either way. To nullify which should be done first (cold stress/heat stress), we gave a gap of 3 months and also insured that there was not much difference exposure temperature in the psychrometric chamber and temperature of ante chamber. These details have been given in the manuscript. The experiment is also designed to align with research work duration of the students.
Reviewer Line 163: “when the caves were in the psychometric chamber itself” → Please correct “caves” to calves and rephrase for clarity: “while the calves remained inside the chamber.”
Authors: Correction has been incorporated.
Reviewer Line 170: “Students with master’s degree...” – consider rephrasing to emphasize technical training, e.g., “Trained personnel holding master’s degrees in Veterinary Physiology collected all data to ensure consistency.”
Authors: Correction has been incorporated.
Reviewer Lines 173–174: Present units consistently in formula definitions. E.g., “Tdb – dry bulb temperature (°C); RH – relative humidity (%)”.
Authors Correction has been incorporated.
Reviewer Line 176: “within the narrow limits (±1.0)” – clarify units: is it ±1.0 °C and ±1.0% RH?
Authors It is ±1.0 °C and ±1.0% RH. Correction has been incorporated.
Reviewer Line 178: “puffed material” is vague – consider specifying insulation material type (e.g., polyurethane foam insulation).
Authors: Suggestion has been incorporated.
Reviewer Consider checking the crude protein value (8.8%) as it seems low for growing calves—please ensure it aligns with NRC recommendations and explain if otherwise.
Authors: Thanks for pointing the mistake. It was a calculation error which has been rectified.
Reviewer Line 190: Please clarify how these values were measured (e.g., data logger, sensors).
Authors: The values were measured using sensors. These sensors are connected with automation panel with both data logger (for storage of data) and also automation facility.
Reviewer Line 195: Define how long RR was counted (e.g., breaths per minute observed over 30 or 60 seconds?).
Authors: It was observed for 1 minute.
Reviewer Line 197–198: Consider mentioning whether thermometers were calibrated, and whether measurements were done by the same person to reduce inter-observer variation.
Authors: All the measurements were done by the same person.
Reviewer Line 202: “least stress to the animals” – suggest rephrasing to “minimizing handling stress.”
Authors: The suggestions have been incorporated.
Line 204: The centrifugation duration and g-force are adequate, but mention the rotor model or radius if available for reproducibility.
Author: The suggestion has been incorporated.
Reviewer Line 217: Clearly state if intra- and inter-assay CVs were verified for each assay in this study.
Authors: The suggestions have been incorporated.
Reviewer Line 219: The URL should not be in the middle of the sentence; cite the software properly with a reference.
Authors: Thanks for suggestion. Changes has been done as suggested.
Reviewer The re-use of the same animals after 3 months introduces potential age-related confounding which is not adequately addressed.
Authors: It took 7 months to complete the experiment. However, in case of Zebu cattle the growth is slow, so the effect on physiological parameters are minimal. Zebu cattle comes to puberty at 22-36 months. Therefore, in present study TNZ will not be significantly affected. Moreover, there could have been individual variation in the calves if new set of animals were taken for second phase of the experiment.
Reviewer Blood sampling time need to be clearly defined.
Authors: Blood samples were taken at 1500 h.
The blood samples were collected at 1500 hours on 10th day of every temperature exposure
Reviewer Lines 236–239 Please clarify whether the breakpoint for pulse rate (PR) was included in the final interpretation of LCT, as it was stated to be “non-significant (p > 0.05)”. It is confusing to include it alongside significant breakpoints. You may consider moving it to a separate sentence or explicitly stating it was excluded from defining TNZ.
Authors: Thanks for pointing it out. After going through the data and final figure, we found that PR had a significant break point. Accordingly, changes have been made.
Reviewer Line 240 Please correct the spacing in “56.73for RR” — a space is needed between the number and the word.
Authors: Correction is incorporated.
Lines 241–244 The phrasing “based on the segmented regression analysis using temperature (21–38°C)” is unclear as 21°C was previously part of the cold exposure phase. Confirm whether this range applies only to the heat stress phase. Additionally, specify why no breakpoint was found for PR.
Authors: Exposure of 21 °C was common for both. Temperature (21–39°C) applies to only the heat stress phase.
A wide variation in PR of the individual animals may be responsible for no breakpoints for PR.
Reviewer Line 246 and Figure 1 Ensure Figure 1 includes clear axes labels, units. Indicate if p-values for each breakpoint are displayed.
Authors: The figure has been checked, it includes the axes labels. X axis and Y axis are same for a particular parameter for both lower temperatures and higher temperature exposure. However, Y axis is different for different parameters. However, if p-values are displayed on figure, that will lead to cluttering and duplication, hence not included.
Reviewer Lines 249–254, Figure 2 please explicitly state that no significant LCT was observed for the erythrocytic parameters (p > 0.05), rather than just saying “not observed”. The term “could not be found” is vague.
Authors: Suggested changes have been incorporated.
Reviewer Lines 259–266, Figure 3 Avoid repeating phrases such as “lower threshold THI for lymphocyte % could not be deciphered (p > 0.05)”. Consider rephrasing to: “No statistically significant lower threshold THI was observed for lymphocyte % (p > 0.05)”.
Authors: Suggested changes have been incorporated.
Reviewer Lines 269–273, Figure 4 The results for liver enzymes should include actual breakpoint temperatures and THI values in the text for clarity, rather than relying solely on figures and tables.
Authors: Please refer to the paragraph below which provides the required information in the original manuscript.
The LCT for AST and ALT activity were observed at 12.30 °C and upper critical UCT for AST and ALT were observed at 30.1 and 34.77 °C (Figure 4) while the lower threshold THI for AST and ALT activity were observed at 54.75 and 54.97, respectively and upper threshold THI at 88.98 and 82.35, respectively (Table 3).
Reviewer Lines 274–278, Figure 5 Sentence construction is unclear: “Similarly, the lower threshold THI was observed at 54.97 and upper threshold THI at 84.29” — please clarify that these values refer to cortisol.
Authors: Suggested changes have been incorporated.
Reviewer Lines 280–285, Figure 6 Please correct formatting errors such as “(p>00.05)” and provide a clearer sentence structure: e.g., “HSP70 did not show statistically significant breakpoints for either LCT or UCT (p > 0.05).”
Authors: Suggested changes have been incorporated.
Reviewer Lines 290–295, Figure 7 Rephrase “did not exhibit (p>0.05) any break point” to a more formal alternative: “no statistically significant breakpoint was observed for TNFα (p > 0.05)”.
Authors: Suggested changes have been incorporated.
Reviewer Line 298–303, Figure 8 The expression “did not show (p<0.05) any UCT” is contradictory. If p < 0.05, a significant result should be present. Confirm and correct — likely it was meant to say “p > 0.05”.
Authors: It is p > 0.05.
Suggested changes have been incorporated.
Reviewer Lines 324–333: Clarify the novelty of the study earlier in the paragraph. Consider stating more directly what is lacking in the literature (e.g., “Chronological evaluation of stress responses in zebu calves has not been reported before.”)
Authors: Suggested changes have been incorporated appropriately.
Reviewer Lines 336–356: The physiological interpretations are reasonable but tend to be speculative in places. Phrases like “in order to maintain sufficient blood flow” and “minimize evaporative heat loss” require stronger citation support or should be softened (e.g., “may suggest an attempt to…”). Also, avoid repetition from the Results.
Authors: As suggested, changes have been incorporated appropriately and few results in discussion have been deleted.
Reviewer Lines 370–379: It would be helpful to elaborate why the THI thresholds are reportedly higher in calves than in lactating cows. Is this purely due to metabolic heat production differences?
Authors: One more reason has been added.
The lower metabolic and fermentative heat production and higher heat loss due to larger surface area per kilogram body weight in calves leads to lower heat load resulting in late initiation of physiological heat loss mechanisms as compared to lactating animals.
Reviewer Lines 390–401: Please clarify if the neutrophil:lymphocyte ratio is explicitly calculated or inferred. The sentence “could have been triggered by increase in cortisol” is speculative; consider rephrasing or supporting with data correlation.
Authors: Neutrophil:Lymphocyte ratio was not calculated, it was inferred.
In the later part of the discussion, it has been discussed, however, speculative phrase has been deleted.
Reviewer Lines 402–423: This section discussing AST and ALT is lengthy and includes repeated information. Streamline to avoid repeating “increased at XX °C and then declined.” Clearly distinguish what is novel here.
Authors: As suggested, many details have been deleted.
Lines 425–440: Provide a clearer conclusion regarding cortisol’s role in early vs. late stress response. Consider adding a sentence to summarize its diagnostic value relative to other markers like RR or HSPs.
Authors: The discussion has been changed as per the suggestions.
Reviewer Lines 441–453: The HSP70 and HSP90 comparison is important, but needs clearer structuring. Present the cold stress findings separately from heat stress findings. Also clarify: how was “better marker” assessed — through earlier breakpoint, magnitude of change, or significance?
Authors: Better marker was assessed based on the earlier break point. Changes have been made as suggested.
Reviewer Lines 455–471: The discussion of variability in HSP expression is relevant but too dense. Condense and emphasize only the most important interpretation points (e.g., HSP90 increased before cortisol in cold stress).
Authors: Changes have been made as suggested.
Reviewer Lines 472–484: Clearly state why TNFα might not respond to cold stress in this context. The current discussion is speculative and somewhat unclear.
Authors: We have inserted that “In present study, no significant change in TNFα reflects a non-inflammatory nature of cold stress and strong adaptive resilience of zebu calves”.
Reviewer Lines 485–493: Consider reducing citations within one sentence. In Line 491, the claim that THI >86 leads to “compromised immune response” needs clearer evidence or a more cautious statement (e.g., “may indicate immune dysregulation”).
Authors: The citation has been reduced and changes has been done as suggested.
Reviewer Lines 505–513: This is an important section summarizing the key stress markers. Clarify how sensitivity was defined (e.g., lowest breakpoint, highest magnitude of change, etc.).
Authors: The sensitivity was defined on the basis lowest breakpoint and it has been inserted in the manuscript.
Reviewer Lines 514–530: This concluding section nicely highlights the novelty of the study, but consider reducing repetition (especially of TNZ ranges). Clarify that your findings provide a multi-parameter TNZ definition, which is a strength.
Authors: The repetition has been deleted. The suggestion about multi-parameter TNZ definition has been incorporated.
Reviewer Lines 532–533: Please rephrase for clarity. Instead of “On the basis of present study, it can be concluded that…,” use a more concise and active structure, e.g., “This study demonstrates that…”
Authors: The suggestion has been incorporated.
Reviewer Line 539: “Were found not be sensitive” is grammatically incorrect. Please correct to “were found not to be sensitive…”
Authors: The correction has been incorporated.
Reviewer Lines 540–541: This comparative sentence is insightful but should be clarified. Suggest rephrasing to: “During heat stress, cortisol elevation appears to precede immune and metabolic responses, whereas in cold stress, these physiological changes occur before an increase in cortisol is observed.”
Authors: The suggestion has been incorporated.
Reviewer Lines 542–544: Consider replacing “approximately” with specific decimal values already reported in the Results for LCT and UCT. This would strengthen the precision of your conclusion.
Authors: The suggestion has been incorporated.
Reviewer Lines 545–548: The final management recommendation is appropriate but would benefit from a more cautious tone. Consider adding: “These findings may help inform thermal comfort and housing strategies in calf management; however, further field validation is recommended.”
Authors: A part of suggestion has been added.
Reviewer Use consistent tense (preferably present simple or present perfect when referring to study findings).
Author: Suggested changes have been done in the conclusion.